# SEPS: Semantic-enhanced Patch Slimming Framework for fine-grained cross-modal alignment

**Xinyu Mao, Junsi Li, Haoji Zhang, Yu Liang, Ming Sun**[*]
School of Computer Science and Engineering
University of Electronic Science and Technology of China
Chengdu,610065, China

## Abstract

Fine-grained cross-modal alignment aims to establish precise local correspondences between vision and language, forming a cornerstone for visual question answering and related multimodal applications. Current approaches face challenges in addressing patch redundancy and ambiguity, which arise from the inherent information density disparities across modalities. Recently, Multimodal Large Language Models (MLLMs) have emerged as promising solutions to bridge this gap through their robust semantic generation capabilities. However, the dense textual outputs from MLLMs may introduce conflicts with the original sparse captions. Furthermore, accurately quantifying semantic relevance between rich visual patches and concise textual descriptions remains a core challenge. To overcome these limitations, we introduce the Semantic-Enhanced Patch Slimming (SEPS) framework, which systematically addresses patch redundancy and ambiguity. Our approach employs a two-stage mechanism to integrate unified semantics from both dense and sparse texts, enabling the identification of salient visual patches. Additionally, it leverages relevance-aware selection with mean value computation to highlight crucial patch-word correspondences, thereby improving cross-modal similarity assessment. Comprehensive experiments on Flickr30K and MS-COCO datasets validate that SEPS achieves superior performance, surpassing existing approaches by 23%-86% in rSum across diverse model architectures, with notable enhancements in text-to-image retrieval scenarios. Our implementation is available at https://github.com/Sweet4tars/seps.git.

## 1 Introduction

Fine-grained cross-modal alignment between vision and language has emerged as a cornerstone for establishing precise local correspondences across modalities, serving as the foundation for visual question answering (Guo et al., 2019), image captioning (Li et al., 2019), and cross-modal retrieval (Fu et al., 2023). As multimodal applications demand increasingly sophisticated understanding capabilities, achieving accurate alignment between visual patches and semantic concepts has become critical for advancing the field.

Despite this importance, existing cross-modal alignment methods universally face the fundamental challenge of bridging the information gap between modalities. This gap stems from the contrasting nature of cross-modal information representation: visual inputs provide dense, continuous spatial information, while textual descriptions offer sparse, discrete semantic anchors that capture only salient scene aspects. With Vision Transformer (ViT) based models (Dosovitskiy et al., 2020) becoming mainstream through efficient patch-based image processing in fine-grained alignment methods, this information gap manifests itself as two problems: patch redundancy, where numerous visual patches contain overlapping or irrelevant information with no explicit textual counterparts, and patch ambiguity, where sparse textual elements are difficult to map reliably to individual patches. These problems particularly impair text-to-image retrieval performance in complex visual scenarios. As

---

[*]Corresponding author:sunm@uestc.edu.cn

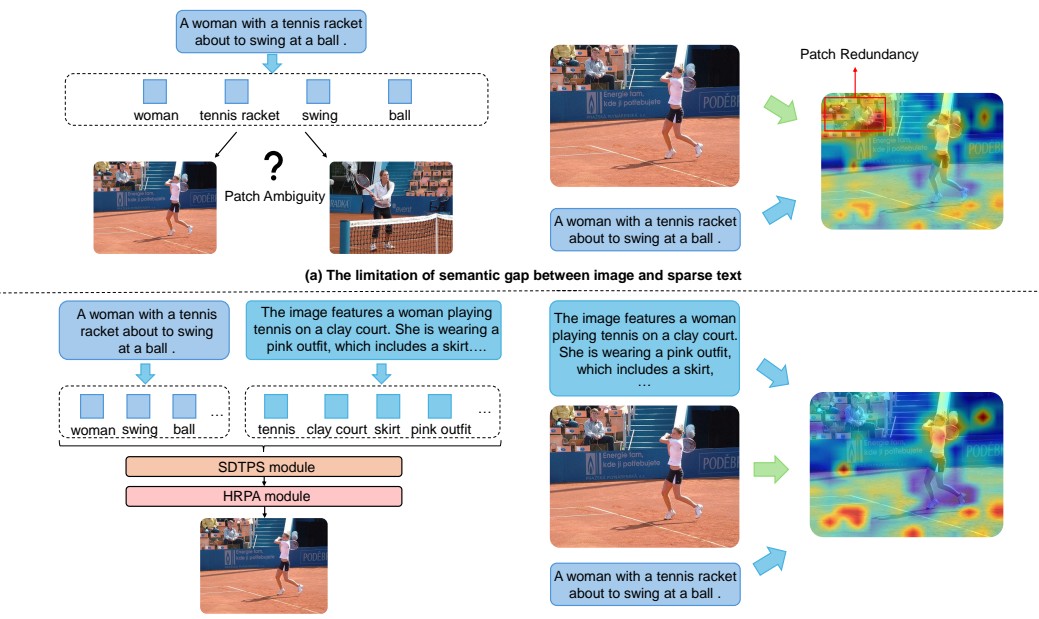

Figure 1: The motivation of our framework, where blue arrows mean language transformer, and green arrows mean vision transformers. (a) Current works suffer from patch ambiguity and patch redundancy due to the limited semantic guidance. (b) Our framework fuses unified semantic derived from dense and sparse texts to guide visual patch selection, and introduces a relevance-aware selection to improve patch-word alignment, which bridges the semantic gap.

illustrated in Figure 1(a), generic captions such as "A woman with a tennis racket about to swing at a ball" lack distinctive visual descriptors, highlighting this inherent information density disparity.

Recently, Multimodal Large Language Models (MLLMs) have emerged as promising solutions to bridge this semantic gap through their robust semantic generation capabilities (Pan et al., 2023; Fu et al., 2024; Liu et al., 2025a). However, MLLM integration introduces semantic inconsistencies, as comprehensive MLLM-generated descriptions may conflict with existing concise captions, potentially causing confusion in cross-modal alignment and degrading retrieval performance. Additionally, accurately quantifying semantic relevance between rich visual patches and concise textual descriptions remains challenging, as conventional alignment methods rely on global averaging, failing to recognize that only a subset of patches are semantically relevant, thus allowing irrelevant regions with low similarity scores to dilute the overall alignment quality.

To overcome these limitations, we introduce the Semantic-Enhanced Patch Slimming (SEPS) framework, which systematically addresses both patch redundancy and ambiguity through strategic MLLM integration, as shown in Figure 1(b). Our key insight centers on employing a two-stage mechanism that integrates unified semantics derived from both dense MLLM-generated texts and sparse original captions, where dense texts provide contextual guidance while sparse texts serve as specific queries for salient patch identification.

Specifically, as illustrated in Figure 2(a), our framework operates through a comprehensive pipeline: we first extract visual patches alongside sparse-text and dense-text feature representations, then aggregate semantically selected visual patches through Sparse and Dense Text-Aware Patch Selection (SDTPS) module, which makes informed selection decisions based on complementary textual perspectives. Finally, we employ our Highly-Relevant Patch-Word Alignment (HRPA) module with relevance-aware selection and mean value computation to facilitate nuanced fine-grained interactions, amplifying highly-relevant patch-word correspondences and improving alignment quality.

The contributions of this paper are as follows:

- To the best of our knowledge, we propose the first systematic framework that strategically employs MLLMs to assist visual patch selection for cross-modal alignment, addressing fundamental patch redundancy and ambiguity challenges.

- We introduce a novel two-stage mechanism that incorporates unified semantic representations derived from both dense and sparse textual modalities. This mechanism eliminates potential semantic inconsistencies, enabling more accurate identification of salient visual patches.

- We develop a relevance-aware selection mechanism augmented by mean value calculation, which enhances the emphasis on critical patch-word correspondences. This design effectively mitigates the averaging bias inherent in traditional methods and improves cross-modal similarity evaluation.

- We achieve superior performance on Flickr30K and MS-COCO datasets, surpassing existing approaches by 23%-86% in rSum across diverse model architectures, with notable enhancements in text-to-image retrieval scenarios.

## 2 RELATED WORK

### 2.1 CROSS-MODAL ALIGNMENT

Cross-modal alignment aims to bridge the semantic gap between vision and language through two primary strategies: coarse-grained and fine-grained alignment. Coarse-grained methods, such as VSE++ (Faghri et al., 2017), compute the global similarity between an entire image and a text. In contrast, fine-grained methods achieve more precise matching by modeling interactions between local features, such as specific image regions and individual words. Early approaches relied on object detectors like Faster R-CNN (Girshick, 2015) to extract visual regions, followed by cross-attention mechanisms for alignment, as seen in SCAN (Lee et al., 2018), SGR (Diao et al., 2021), and CHAN (Pan et al., 2023). However, this paradigm is computationally expensive, its performance is dependent on the detector's accuracy, and it is prone to error propagation. Recently, end-to-end models based on the Vision Transformer (ViT) (Dosovitskiy et al., 2020) have become mainstream. ViT processes images by dividing them into patches, but this introduces new challenges: patch redundancy and patch ambiguity. To mitigate these problems, recent work like LAPS (Fu et al., 2024) demonstrates the potential of using linguistic supervision to prune redundant patches by leveraging captions from standard datasets. While effective, the semantic sparsity inherent in these captions creates a performance bottleneck, particularly in complex visual scenarios. Building on this foundation, our work addresses this bottleneck by integrating dense semantic guidance with the original sparse text, exploring how this hybrid supervision can unlock further improvements in patch selection.

### 2.2 INFORMATION DENSITY AND DENSE TEXT SUPERVISION

Visual signals are dense, while textual descriptions are relatively sparse, leading to a fundamental information capacity mismatch. This challenge has motivated a significant recent trend: the use of Multimodal Large Language Models (MLLMs) to generate dense text, which provides a much richer supervisory signal to bridge this gap. Several works have begun to leverage this rich data. One line of research focuses on enhancing a model's general long-text capabilities through pre-training, such as in LongCLIP (Zhang et al., 2024) and LoTLIP (Wu et al., 2024). Other approaches tackle the density mismatch at the feature representation level. For instance, methods learn diverse embedding sets (e.g. PCME (Chun et al., 2021)) or design asymmetric architectures (e.g. AVSE (Liu et al., 2025c)) to accommodate modal differences, while the prominent D2S-VSE (Liu et al., 2025b) uses dense text as a "teacher" signal to distill knowledge and enrich sparse text representations.

However, a common thread unites these existing methods: they primarily optimize at the feature representation or alignment stage—either by improving the model's global understanding or by enhancing the textual representations. The granular semantic details within dense text have not been exploited to directly address visual information redundancy at the input level. Therefore, our work explores how to leverage this fine-grained information to directly guide the visual patch selection process, aiming to solve the information mismatch by proactively refining the visual input itself.

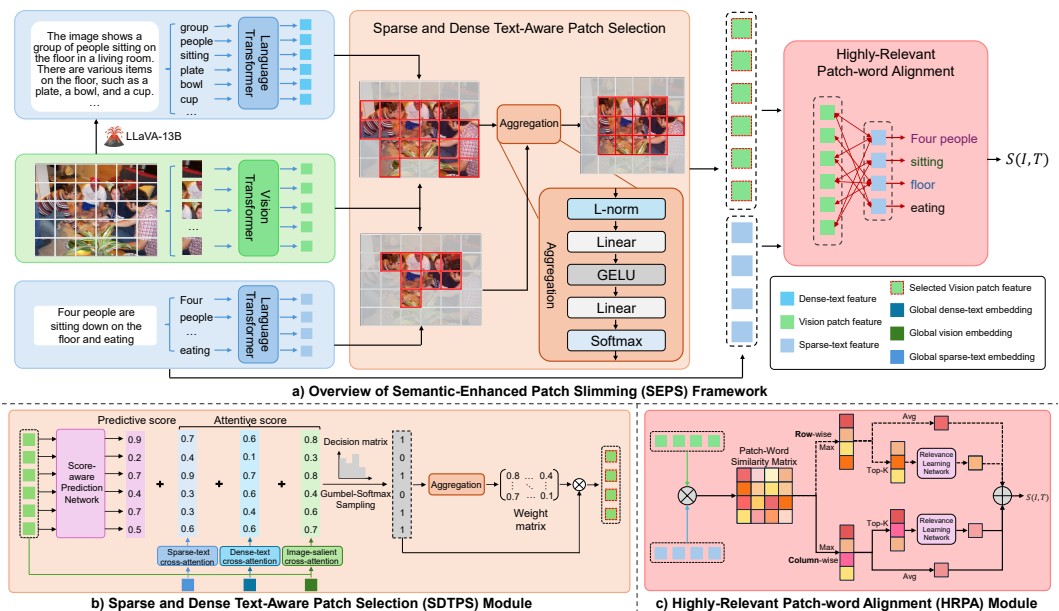

Figure 2: (a) Overview of our Semantic-Enhanced Patch Slimming(SEPS) Framework for fine-grained cross-modal alignment. Given an image-text pair $(I, T)$, we first use pure Transformer encoders to extract visual patch features and textual word features. Then, we propose the SDTPS module to identify text-relevant patches under the support of dense text generated by MLLMs. Finally, we propose the HRPA module to compute the patch-word alignment score $S(I, T)$. (b)(c) The detailed architecture of the proposed SDTPS and HRPA modules, respectively.

## 3 METHODOLOGY

This section introduces the SEPS framework, which integrates dense textual representations generated by MLLMs with original sparse textual features through a novel two-stage mechanism. Coupled with a relevance-aware mechanism, the framework effectively addresses semantic inconsistency in large model integration and similarity shift bias in existing alignment computation, thereby bridging the semantic gap between visual and textual modalities and resolving patch redundancy and ambiguity issues prevalent in current alignment methodologies. The framework architecture encompasses three principal modules: the dense text generation component detailed in Section 3.1, the Sparse and Dense Text-Aware Patch Selection Module presented in Section 3.2 and visualized in Figure 2(b), and the Highly-relevant Patch-word Alignment module described in Section 3.3 and illustrated in Figure 2(c).

### 3.1 DENSE TEXT GENERATION BASED ON MLLMS

To generate dense textual representations for visual inputs, we employ the pre-trained multimodal model LLaVa (Liu et al., 2023). For a given input image $I$, we utilize LLaVa with the instructional prompt *"Provide a comprehensive description of this image"*. The model subsequently generates semantically dense textual output that encodes granular visual information, leveraging its enhanced visual perception capabilities. Implementation details and parameter configurations for LLaVa are specified in Section 4.3. This methodology ensures that the linguistic representation maintains informational richness comparable to the visual modality, thereby mitigating the cross-modal information asymmetry inherent in traditional multimodal systems.

### 3.2 SPARSE AND DENSE TEXT-AWARE PATCH SELECTION MODULE

The SDTPS module employs a two-stage mechanism to fuse unified semantics derived from dense and sparse texts, thereby identifying visual patches that exhibit robust semantic alignment with such integrated semantic representations. In the first stage, semantic scoring assigns each visual patch a comprehensive score derived from multiple information sources, including sparse textual features,

dense textual features, and intrinsic image characteristics. Subsequently, the second stage employs a decision and aggregation process that utilizes a learned weight matrix to identify and extract the most semantically relevant patches for further alignment.

### 3.2.1 SEMANTIC SCORING

In first stage, the module employs a score-aware prediction network to assess the semantic relevance of individual visual patches. This network predicts the final scores that each visual patch would obtain from three distinct cross-attention mechanisms, thereby improving the learning capacity of the semantic scoring stage. The network comprises a two-layer MLP followed by a sigmoid activation function.

$$s_i^p = \sigma\left(\text{MLP}\left(\boldsymbol{v}_i\right)\right), \, i \in \{1, \ldots, N\}, \tag{1}$$

where $s_i^p \in [0, 1]$ represents the significance score for the $i$-th patch, and $v_i \in V = \{v_1, v_2, \ldots, v_N\}$ denotes the visual patch feature vector, $\sigma$ means sigmoid activation function. A higher value of $s_i^p$ indicates greater semantic significance of the patch $v_i$.

This prediction network is primarily integrated with attention scores derived from textual content and image self-attention mechanisms to achieve better cross-modal alignment (Meng et al., 2022; Rao et al., 2021). However, given that most text in existing datasets is sparse, an information capacity gap emerges between visual and textual modalities. To address this limitation, we leverage dense text generated by MLLMs to enhance the textual relevance of the significance scores $s_i^p$ for visual patches.

Building upon the cross-attention between visual patches and sparse textual representations, we propose an additional attention mechanism: cross-attention between visual patches and dense textual representations. Therefore, the complete attention score computation formula is as follows:

$$s_i^{st} = \text{Norm}(v_i^T \cdot E_{st}/d) \quad s_i^{dt} = \text{Norm}(v_i^T \cdot E_{dt}/d) \quad s_i^{im} = \text{Norm}(v_i^T \cdot E_{im}/d), \tag{2}$$

where $s_i^{st}$ represents the sparse-text relevance of the visual patch, $s_i^{dt}$ denotes the dense-text relevance of the visual patch, and $s_i^{im}$ represents the significance of the $i$-th patch in the visual dimension. Norm represents the normalization of attention scores into the $[0, 1]$ range to maintain consistency with the outputs of the prediction network $s_i^p$. $E_{st}$, $E_{dt}$, and $E_{im}$ denote the global embedding vectors of sparse text, dense text, and image, respectively. $d$ is the number of embedding dimensions. Finally the total significance score formula with $\beta$ as a weight parameter is as follows,:

$$s_i = (1 - 2\beta) \cdot s_i^p + \beta \cdot (s_i^{st} + s_i^{dt} + 2s_i^{im}) \tag{3}$$

### 3.2.2 DECISION AND AGGREGATION

In second stage, the computed significance scores $s = [s_1, s_2, \ldots, s_N] \in \mathbb{R}^N$ undergo a binary mapping process through a differentiable decision matrix. Compared to naive sampling approaches, such as selecting the top-K patches, the Gumbel-Softmax technique provides smooth and differentiable sampling capabilities(Maddison et al., 2016). Based on this technique, we follow the previous sampling methodology to obtain differentiable decision matrices $D_s$ and $D_d$ for sparse-text and dense-text, respectively (Fu et al., 2024). $D$ is a one-hot matrix where $'1'$ indicates a significant patch and $'0'$ indicates a redundant patch. Based on the sparse-text matrix $D_s$ and dense-text matrix $D_d$, we can select significant patches $V_s = \{v_1^s, v_2^s, \ldots, v_{N_s}^s\} \in \mathbb{R}^{N_s \times d}$ for sparse-text and $V_d = \{v_1^d, v_2^d, \ldots, v_{N_d}^d\} \in \mathbb{R}^{N_d \times d}$ for dense-text.

These binary decisions are subsequently processed through an aggregation network that learns multiple aggregation weights (Zong et al., 2022) and aggregates $N_s$ and $N_d$ significant patches to generate $N_c$ informative patches.

$$\hat{v}_j = \sum_{i=1}^{N_s}(W_s)_{ij} \cdot v_i^s + \sum_{i=1}^{N_d}(W_d)_{ij} \cdot v_i^d, \quad j \in \{1, \ldots, N_c\} \tag{4}$$

where $(W_s)_{ij}$ and $(W_d)_{ij}$ are the elements of the normalized weight matrices $W_s \in \mathbb{R}^{N_s \times N_c}$ and $W_d \in \mathbb{R}^{N_d \times N_c}$. $N_c$ is the number of aggregated patches ($N_c < \max(N_s, N_d)$), and we have $\sum_{i=1}^{N_s}(W_s)_{ij} = 1$ and $\sum_{i=1}^{N_d}(W_d)_{ij} = 1$. The weight matrices $W_s$ and $W_d$ are learned by an MLP

followed by a softmax function, taking significant patches based on sparse-text and dense-text as input, respective: $W_s = \text{Softmax}(\text{MLP}(V_s))$ and $W_d = \text{Softmax}(\text{MLP}(V_d))$.

Specifically, we treat the decision matrices $D_s$ and $D_d$ as mask matrices to select the significant patch features $V_s$ and $V_d$ before computing the softmax function. The aggregation network can adaptively aggregate patches with similar semantics and is differentiable for end-to-end training.

### 3.3 HIGHLY-RELEVANT PATCH-WORD ALIGNMENT

The HRPA module introduces relevance-aware selection with mean value computation to facilitate nuanced fine-grained interactions, amplifying highly-relevant patch-word correspondences. As shown in Figure.2(c), we compute the fine-grained alignment by the set of selected visual patches $\hat{V}$ and initial sparse textual words $T$. For convenience, we approximate that $|\hat{V}| = N_c, |T| = M$. We first calculate the token-wise similarity to generate the patch-word similarity matrix $A \in \mathbb{R}^{N_c \times M}$, where $A_{ij} = \frac{(\hat{v}_i)^T t_j}{\|\hat{v}_i\| \|t_j\|}$ represents the alignment score between the $i$-th visual patch and the $j$-th textual word.

Next, we employ a relevance-aware selection to aggregate the cross-modal alignment, which enhances the contribution of maximally relevant patch-word pairs to image-text similarity, thereby improving alignment quality. We identify the most aligned textual token (or visual patch) for each visual patch (or textual token), and use the relevant learning network to transform the selected maximum scores into a scalar value. Then calculate the average of total aligned scores. The sum of these two values represent the overall alignment score between the image $I$ and the sentence $T$, denoted $S(I, T)$.

$$S(I,T) = \underbrace{\left(\frac{1}{N_c} \sum_{i=1}^{N_c} \max_j (A)_{ij} + \text{MLP}(\text{TOPK}(\max_j (A)_{ij}))\right.}_{\text{patch-to-word alignment}}$$
$$\underbrace{\left. + \frac{1}{M} \sum_{j=1}^{M} \max_i (A)_{ij} + \text{MLP}(\text{TOPK}(\max_i (A)_{ij}))\right)}_{\text{word-to-patch alignment}} \tag{5}$$

Following prior work, we adopt a bidirectional triplet loss with hard negative mining(Faghri et al., 2017):

$$\mathcal{L}_{\text{align}} = \sum_{(I,T)} \left(\left[\alpha - S(I,T) + S(I,\hat{T})\right]_+ \right.$$
$$\left. + \left[\alpha - S(I,T) + S(\hat{I},T)\right]_+\right) \tag{6}$$

where $\alpha$ is the margin, $[x]_+ = \max(x, 0)$, and $(I, T)$ denotes a positive image–text pair within the mini-batch. The hardest negatives are defined as $\hat{T} = \arg\max_{j \neq T} S(I, j)$ and $\hat{I} = \arg\max\_i \neq IS(i, T)$ for text and image, respectively.

Furthermore, to enhance training stability, we constrain the proportion of selected patches to a target value $\rho$(Rao et al., 2021), and supervise this constraint using mean-squared-error losses computed from the sparse-text and dense-text views, respectively. Finally, we combine the cross-modal alignment loss $\mathcal{L}_{\text{align}}$ Eq.6 with the ratio constraint loss $\mathcal{L}_{\text{ratio}}$:

$$\mathcal{L}_{\text{ratio}} = \left(\rho - \lambda_1 \cdot \frac{1}{N_s} \sum_{i=1}^{N_s} (D_s)_i - \lambda_2 \cdot \frac{1}{N_d} \sum_{i=1}^{N_d} (D_d)_i\right)^2,$$
$$\mathcal{L} = \mathcal{L}_{\text{align}} + \mathcal{L}_{\text{ratio}} \tag{7}$$

where $\lambda_1$ and $\lambda_2$ are constant coefficients for sparse text and dense text.

Table 1: Comparisons of image-text retrieval performances on Flickr30K and MS-COCO test-set. We list the details of feature encoding, image resolution, and the number of obtained regions/patches by visual encoder (e.g. "ViT-Base-224" represents the base-version of Vision Transformer with 224×224 image resolution input, regarding 16×16 pixels as one patch, and getting 14×14 visual patches for one image). FG indicates whether it is the fine-grained cross-modal alignment. The best results are marked **bold**, and the second best results are marked underline.

| Method | FG | Flickr30K 1K Image-to-Text R@1 | R@5 | R@10 | Text-to-Image R@1 | R@5 | R@10 | rSum | MS-COCO 1K Image-to-Text R@1 | R@5 | R@10 | Text-to-Image R@1 | R@5 | R@10 | rSum | MS-COCO 5K Image-to-Text R@1 | R@5 | R@10 | Text-to-Image R@1 | R@5 | R@10 | rSum |
|---|---|---|---|---|---|---|---|---|---|---|---|---|---|---|---|---|---|---|---|---|---|---|
| *ViT-Base-224 + BERT-base, 14×14 patches* | | | | | | | | | | | | | | | | | | | | | | |
| VSE++ (Faghri et al., 2017) | ✗ | 71.8 | 92.8 | 96.5 | 59.4 | 84.7 | 90.9 | 496.1 | 75.0 | 94.6 | 98.0 | 62.7 | 89.4 | 94.9 | 514.6 | 52.4 | 80.3 | 88.8 | 40.6 | 70.4 | 81.1 | 413.4 |
| SCAN (Lee et al., 2018) | ✓ | 69.5 | 90.9 | 95.6 | 56.4 | 83.1 | 90.0 | 485.6 | 76.0 | 95.4 | 98.1 | 64.5 | 90.8 | 95.8 | 520.6 | 53.9 | 81.8 | 90.0 | 42.9 | 72.3 | 82.5 | 423.5 |
| SGR (Diao et al., 2021) | ✓ | 69.7 | 90.8 | 95.2 | 59.1 | 84.1 | 89.9 | 488.7 | 77.2 | 95.0 | 98.0 | 65.1 | 90.7 | 95.8 | 521.8 | 54.9 | 82.8 | 90.5 | 42.8 | 72.2 | 82.5 | 425.8 |
| CHAN (Pan et al., 2023) | ✓ | 69.2 | 91.8 | 95.0 | 58.4 | 84.9 | 90.6 | 489.9 | 77.1 | 95.1 | 98.1 | 65.0 | 91.0 | 96.0 | 522.2 | 56.3 | 83.2 | 90.1 | 43.0 | 72.6 | 82.8 | 428.0 |
| LAPS (Fu et al., 2024) | ✓ | 74.0 | 93.4 | 97.4 | 62.5 | 87.3 | 92.7 | 507.3 | 78.7 | 95.5 | 98.3 | 66.2 | 91.3 | 96.2 | 526.3 | 57.5 | 84.0 | 90.8 | 44.5 | 74.0 | 83.6 | 434.4 |
| AVSE (Liu et al., 2025c) | ✗ | 76.0 | 94.6 | 97.5 | 62.7 | 88.4 | 93.1 | 512.3 | 79.8 | 95.6 | 98.3 | 67.0 | 91.5 | 96.3 | 528.5 | 58.8 | 84.3 | 91.0 | 45.1 | 74.3 | 83.9 | 437.4 |
| D2S-VSE (Liu et al., 2025b) | ✗ | 82.8 | **96.1** | **98.3** | 68.5 | 91.3 | 94.9 | 531.9 | 80.1 | **97.0** | **99.2** | 68.1 | 92.5 | 96.7 | 533.7 | 60.1 | **85.5** | **92.5** | 46.3 | 75.9 | 85.2 | 445.6 |
| **SEPS** | ✓ | **86.1** | 93.7 | 96.9 | **86.9** | **98.1** | **99.2** | **560.9** | **89.0** | 94.8 | 98.0 | **88.5** | **99.3** | **99.8** | **569.5** | **73.9** | 85.2 | 92.1 | **73.5** | **94.5** | **97.8** | **516.9** |
| *ViT-Base-384 + BERT-base, 24×24 patches* | | | | | | | | | | | | | | | | | | | | | | |
| VSE++ (Faghri et al., 2017) | ✗ | 77.1 | 95.7 | 97.5 | 65.8 | 90.2 | 94.3 | 520.5 | 77.0 | 95.7 | 98.4 | 64.6 | 91.1 | 96.2 | 523.0 | 54.9 | 82.8 | 90.4 | 42.4 | 72.4 | 82.8 | 425.8 |
| SCAN (Lee et al., 2018) | ✓ | 75.4 | 94.4 | 96.9 | 63.6 | 88.6 | 93.5 | 512.5 | 76.1 | 95.5 | 98.5 | 65.1 | 91.6 | 96.3 | 523.1 | 53.3 | 81.8 | 90.0 | 42.6 | 72.6 | 82.9 | 423.1 |
| SGR (Diao et al., 2021) | ✓ | 76.9 | 94.9 | 98.1 | 64.2 | 88.4 | 93.3 | 515.8 | 75.8 | 95.7 | 98.6 | 65.6 | 92.0 | 96.5 | 524.2 | 53.3 | 81.0 | 89.6 | 42.9 | 73.1 | 83.7 | 423.6 |
| CHAN (Pan et al., 2023) | ✓ | 75.4 | 94.5 | 97.6 | 63.2 | 88.6 | 93.1 | 512.4 | 78.1 | 95.8 | 98.6 | 66.1 | 92.1 | 96.6 | 527.3 | 55.6 | 83.8 | 91.2 | 43.4 | 73.6 | 83.5 | 431.1 |
| LAPS (Fu et al., 2024) | ✓ | 79.0 | 96.0 | 98.1 | 67.3 | 90.5 | 94.5 | 525.4 | 78.6 | 96.3 | 98.9 | 68.0 | 92.4 | 96.8 | 531.0 | 57.4 | 84.9 | 92.5 | 46.4 | 75.8 | 85.2 | 442.2 |
| AVSE (Liu et al., 2025c) | ✗ | 80.3 | 96.4 | 98.7 | 67.9 | 91.2 | 94.7 | 529.2 | 81.1 | 97.1 | 99.0 | 68.3 | 92.7 | 97.0 | 535.2 | 61.2 | 86.3 | 93.2 | 46.2 | 75.9 | 85.0 | 448.3 |
| D2S-VSE (Liu et al., 2025b) | ✗ | 84.1 | **97.5** | **99.1** | 70.3 | 91.6 | 95.3 | 537.9 | 80.8 | **97.2** | **99.1** | 69.0 | 92.9 | 96.8 | 535.8 | 60.6 | 86.5 | 93.2 | 46.8 | 76.4 | 85.7 | 449.1 |
| **SEPS** | ✓ | **90.7** | 94.4 | 98.4 | **89.3** | **99.3** | **99.5** | **571.5** | **90.9** | 96.1 | 98.8 | **91.0** | **99.5** | **99.8** | **576.1** | **77.8** | **88.7** | **94.8** | **78.5** | **96.3** | **98.7** | **534.6** |
| *Swin-Base-224 + BERT-base, 7×7 patches* | | | | | | | | | | | | | | | | | | | | | | |
| VSE++ (Faghri et al., 2017) | ✗ | 82.5 | 96.5 | 98.9 | 70.0 | 91.4 | 95.1 | 534.4 | 83.3 | 97.5 | 99.3 | 71.0 | 93.0 | 96.7 | 540.9 | 64.0 | 88.2 | 94.2 | 49.9 | 78.0 | 86.6 | 460.9 |
| SCAN (Lee et al., 2018) | ✓ | 79.0 | 95.9 | 98.2 | 67.7 | 90.6 | 94.9 | 526.3 | 80.9 | 97.0 | 99.1 | 69.7 | 93.1 | 97.1 | 536.9 | 60.7 | 86.6 | 93.2 | 48.1 | 77.1 | 86.1 | 451.8 |
| SGR (Diao et al., 2021) | ✓ | 80.4 | 97.0 | 98.7 | 66.9 | 90.2 | 94.5 | 527.6 | 81.2 | 97.1 | 99.1 | 69.9 | 93.2 | 97.2 | 537.7 | 61.0 | 86.7 | 93.2 | 48.6 | 77.2 | 86.3 | 453.1 |
| CHAN (Pan et al., 2023) | ✓ | 81.4 | 97.0 | 98.6 | 68.5 | 90.6 | 94.5 | 530.6 | 81.6 | 97.2 | 99.3 | 70.6 | 93.7 | 97.6 | 539.8 | 64.1 | 87.9 | 93.5 | 49.1 | 77.3 | 86.1 | 458.0 |
| LAPS (Fu et al., 2024) | ✓ | 82.4 | 97.4 | 99.5 | 70.0 | 91.7 | 95.4 | 536.3 | 84.0 | 97.6 | 99.3 | 72.1 | 93.7 | 97.4 | 544.1 | 64.5 | 89.2 | 94.4 | 51.6 | 78.9 | 87.2 | 465.8 |
| AVSE (Liu et al., 2025c) | ✗ | 83.9 | 97.4 | 99.4 | 70.0 | 92.4 | 95.6 | 538.7 | 84.9 | **98.0** | 99.3 | 72.1 | 94.0 | 97.4 | 545.7 | 66.2 | **89.8** | **94.7** | 51.7 | 79.2 | 87.3 | 468.9 |
| D2S-VSE (Liu et al., 2025b) | ✗ | 87.2 | **98.4** | **99.9** | 73.0 | 93.5 | 96.7 | 548.7 | 82.4 | 97.6 | 99.3 | 70.3 | 93.7 | 97.4 | 540.7 | 63.9 | 87.7 | 94.0 | 49.3 | 78.3 | 87.2 | 460.4 |
| **SEPS** | ✓ | **89.8** | 96.9 | 98.7 | **88.0** | **98.9** | **99.6** | **572.0** | **87.2** | 94.9 | 98.3 | **84.7** | **99.0** | **99.8** | **563.9** | **71.9** | 86.0 | 92.4 | **66.8** | **92.2** | **96.8** | **506.1** |
| *Swin-Base-384 + BERT-base, 12×12 patches* | | | | | | | | | | | | | | | | | | | | | | |
| VSE++ (Faghri et al., 2017) | ✗ | 83.8 | 97.5 | 99.2 | 71.1 | 93.2 | 96.2 | 540.6 | 82.9 | 97.7 | 99.4 | 71.3 | 93.5 | 97.3 | 542.1 | 63.0 | 88.5 | 94.3 | 50.1 | 78.9 | 87.4 | 462.2 |
| SCAN (Lee et al., 2018) | ✓ | 81.9 | 96.9 | 98.9 | 70.0 | 92.7 | 95.8 | 536.1 | 81.6 | 96.8 | 99.1 | 69.1 | 92.7 | 96.7 | 536.1 | 61.1 | 87.3 | 93.3 | 47.8 | 76.9 | 85.9 | 452.4 |
| SGR (Diao et al., 2021) | ✓ | 80.7 | 96.8 | 99.0 | 69.9 | 91.7 | 95.3 | 533.4 | 81.9 | 96.7 | 99.1 | 69.3 | 92.8 | 96.7 | 536.6 | 62.8 | 87.0 | 92.9 | 48.1 | 77.0 | 86.0 | 453.8 |
| CHAN (Pan et al., 2023) | ✓ | 81.2 | 96.7 | 98.8 | 70.3 | 92.2 | 95.9 | 535.0 | 83.1 | 97.3 | 99.2 | 70.4 | 93.1 | 97.1 | 540.2 | 63.4 | 88.4 | 94.1 | 49.2 | 77.9 | 86.6 | 459.5 |
| LAPS (Fu et al., 2024) | ✓ | 85.1 | 97.7 | 99.2 | 74.0 | 93.0 | 96.3 | 545.3 | 84.1 | 97.4 | 99.2 | 72.1 | 93.9 | 97.4 | 544.1 | 67.1 | 88.6 | 94.3 | 53.0 | 79.5 | 87.6 | 470.1 |
| AVSE (Liu et al., 2025c) | ✗ | 87.1 | 98.3 | 99.2 | 73.6 | 93.5 | 96.5 | 548.2 | 85.1 | **98.2** | **99.5** | 71.6 | 94.0 | 97.5 | 545.9 | 68.6 | **90.2** | **95.6** | 52.2 | 79.6 | 87.8 | 474.0 |
| D2S-VSE (Liu et al., 2025b) | ✗ | 87.8 | **99.0** | **99.7** | 75.7 | 94.1 | 96.9 | 553.2 | 83.8 | 97.9 | 99.4 | 71.9 | 94.2 | 97.9 | 544.7 | 65.2 | 89.2 | 94.6 | 51.3 | 79.4 | 87.9 | 467.7 |
| **SEPS** | ✓ | **93.6** | 98.3 | 99.2 | **91.6** | **99.4** | **99.8** | **581.9** | **89.5** | 96.5 | 99.0 | **87.1** | **99.2** | **99.9** | **571.2** | **74.7** | 88.4 | 94.3 | **70.3** | **93.8** | **97.6** | **519.1** |

Figure 3: The retrieval performance of different selection ratios $\rho$, constant coefficients $\lambda_1$ and $\lambda_2$ with various visual encoders on Flickr30K.

# 4 EXPERIMENTS

## 4.1 DATASETS

Following prior works (Diao et al., 2021; Faghri et al., 2017; Lee et al., 2018), we evaluate our model on the widely-used Flickr30K (Young et al., 2014) and MS-COCO (Lin et al., 2014) benchmarks. Each image in these datasets is paired with five textual captions. For Flickr30K, we adopt the standard split of 29,000 training, 1,000 validation, and 1,014 test images. For MS-COCO, we use the common split 113,287 for training, 5,000 for validation, and 5,000 for testing. We report results on both the 1K test set (averaged over 5 folds) and the full 5K test set.

Table 2: The comparisons of image-text retrieval for Vision-Language Pre-training (VLP) Models. *FG* indicates whether the method fine-grained alignment. * means the zero shot learning.

| Method | FG | Flickr30K 1K | | | | MS-COCO 5K | | | |
| | | Image-to-Text | | Text-to-Image | | Image-to-Text | | Text-to-Image | |
| | | R@1 | R@5 | R@1 | R@5 | R@1 | R@5 | R@1 | R@5 |
| *CLIP-ViT-Base-224 + CLIP-BERT-Base, $14\times14$ patches* | | | | | | | | | |
| CLIP* (Radford et al., 2021) | ✗ | 81.4 | 96.2 | 61.1 | 85.4 | 52.3 | 76.2 | 33.3 | 58.2 |
| VSE++ (Faghri et al., 2017) | ✗ | 92.2 | 99.1 | 80.5 | 95.6 | 68.0 | 88.2 | 53.6 | 79.7 |
| SCAN (Lee et al., 2018) | ✓ | 88.2 | 98.1 | 75.3 | 93.1 | 65.4 | 88.0 | 50.7 | 77.6 |
| LAPS (Fu et al., 2024) | ✓ | 92.9 | **99.3** | 80.6 | 95.5 | 69.8 | 90.4 | 54.3 | 80.0 |
| **SEPS** | ✓ | **94.7** | 97.6 | **93.1** | **97.7** | **84.1** | **91.2** | **78.4** | **95.5** |
| *CLIP-ViT-Large-224 + CLIP-BERT-Large, $16\times16$ patches* | | | | | | | | | |
| CLIP* (Radford et al., 2021) | ✗ | 85.0 | 97.7 | 61.3 | 87.0 | 55.9 | 79.1 | 35.9 | 60.9 |
| VSE++ (Faghri et al., 2017) | ✗ | 94.0 | 99.5 | 83.4 | 96.4 | 68.5 | 89.4 | 56.7 | 81.9 |
| SCAN (Lee et al., 2018) | ✓ | 90.0 | 98.5 | 82.0 | 95.9 | 68.0 | 90.4 | 53.2 | 80.7 |
| LAPS (Fu et al., 2024) | ✓ | 94.6 | **99.9** | 84.9 | 97.3 | 72.9 | **91.7** | 57.1 | 81.3 |
| **SEPS** | ✓ | **95.8** | 98.4 | **95.1** | **98.1** | **86.5** | 91.7 | **79.3** | **95.8** |

## 4.2 METRICS

We adopt Recall@K (R@K, $K \in \{1, 5, 10\}$) and rSum as evaluation metrics. R@K measures the percentage of ground truth in the retrieved top-K lists, while rSum aggregates multiple R@K in both directions (image-to-text and text-to-image) to summarize overall retrieval quality.

## 4.3 IMPLEMENTATION DETAILS

Our code is based on the public code of LAPS (Fu et al., 2024). For dense text generation, we use LLaVa (Liu et al., 2023) to produce detailed textual descriptions. The generation process was configured with a Top-P of 0.9, a temperature of 0.2, and a limit of 500 new tokens. Notably, we treat dense-text generation as an image preprocessing step with no gradients backward. Therefore, no information leakage occurs from the test datasets. For vision encoder, we adopt base-size Vision Transformer (ViT) (Dosovitskiy et al., 2020) (a patch is $16\times16$ pixels) and Swin Transformer (Swin) (Liu et al., 2021) (a patch is $32\times32$ pixels). Images are resized to $224\times224$ or $384\times384$, yielding $14\times14$ and $24\times24$ patch grids for ViT and $7\times7$ and $12\times12$ for Swin. Text is encoded with base-size BERT (Devlin et al., 2019). The whole framework is trained for 30 epochs using AdamW (Loshchilov & Hutter, 2017) optimizer with a batch size of 32 and an initial learning rate of 1e-4. We use loss with margin $\alpha = 0.2$ and set the constant coefficients $\lambda_1 = \lambda_2 = 1$. For sparsification, we adopt fixed ratios by default: on ViT, selection $\rho = 0.5$; on Swin, selection $\rho = 0.8$.

## 4.4 COMPARISON WITH STATE-OF-THE-ART METHODS

Following the standard protocols of two benchmarks (Faghri et al., 2017; Zhang et al., 2022), we systematically compare the retrieval performance of SEPS with recent state-of-the-art methods on Flickr30K and MS-COCO. Table 1 details the feature encoders, input resolutions, and whether fine-grained alignment (FG) was adopted for each method. The performance of competing methods is reported directly from their original publications, supplementing with ensemble versions where necessary for comparison. Firstly, we introduce four SOTA cross-modal alignment methods:

- **CHAN (Pan et al., 2023):** Applies a hard-coded selection strategy atop the foundational fine-grained alignment SCAN (Lee et al., 2018), retaining the maximum cross-attention alignment scores.

- **LAPS (Fu et al., 2024):** A fine-grained approach that prunes redundant patches under language guidance, followed by semantic and spatial calibration to enable sparse, bidirectional patch–word alignment.

Table 3: The zero-shot evaluation on image-text retrieval task. All models are trained by CLIP backbones of ViT-B/16 in Flickr dataset($\#$ is untrained), and evaluated in MS-COCO dataset.

| Method | FG | MS-COCO 1K | | | | MS-COCO 5K | | | |
| | | Image-to-Text | | Text-to-Image | | Image-to-Text | | Text-to-Image | |
| | | R@1 | R@5 | R@1 | R@5 | R@1 | R@5 | R@1 | R@5 |
|---|---|---|---|---|---|---|---|---|---|
| CLIP$^{\#}$ (Radford et al., 2021) | ✗ | - | - | - | - | **52.3** | **76.2** | 33.3 | 58.2 |
| VSE++ (Faghri et al., 2017) | ✗ | 28.1 | 56.4 | 20.5 | 46.8 | 13.2 | 30.5 | 9.1 | 24.6 |
| D2S-VSE (Liu et al., 2025b) | ✗ | 32.3 | 62.3 | 24.8 | 53.9 | 15.9 | 34.7 | 11.3 | 28.3 |
| SCAN (Lee et al., 2018) | ✓ | 30.5 | 59.8 | 23.2 | 51.5 | 14.8 | 32.9 | 10.5 | 26.8 |
| LAPS (Fu et al., 2024) | ✓ | 47.4 | 74.9 | 35.8 | 66.1 | 27.1 | 50.5 | 19.0 | 50.5 |
| **SEPS** | ✓ | **65.8** | **77.4** | **62.8** | **87.5** | 45.8 | 60.2 | **44.3** | **69.5** |

Table 4: The zero-shot evaluation on visual grounding task. All models are trained by CLIP backbones of ViT-B/16 in Flickr dataset ($\#$ is untrained). Following ReCLIP (Subramanian et al., 2022), we apply the Grad-GAM (Selvaraju et al., 2020) to select the bounding box from proposals.

| Models | FG | RefCOCO | | | RefCOCO+ | | | RefCOCOg | |
| | | Val | TestA | TestB | Val | TestA | TestB | Val | Test |
|---|---|---|---|---|---|---|---|---|---|
| CLIP$^{\#}$ (Radford et al., 2021) | ✗ | 39.3 | 45.3 | 34.2 | 41.2 | 47.0 | 36.8 | 45.0 | 45.9 |
| VSE++ (Faghri et al., 2017) | ✗ | 40.7 | 46.3 | 33.6 | 43.2 | 49.0 | 35.6 | 44.2 | 43.9 |
| SCAN (Lee et al., 2018) | ✓ | 41.8 | 47.3 | 44.4 | 43.2 | 49.3 | 36.8 | 45.2 | 46.0 |
| LAPS (Fu et al., 2024) | ✓ | 44.2 | 49.9 | 38.4 | 46.7 | 52.3 | 41.6 | 51.3 | 51.2 |
| **SEPS** | ✓ | **48.7** | **52.3** | **43.4** | **51.2** | **54.8** | **46.6** | **55.3** | **55.2** |

- **AVSE (Liu et al., 2025c):** A coarse-grained approach that constructs multi-view global image embeddings via radial-biased sampling and performs Asymmetric Embedding Optimal Matching (AEOM) for global alignment.

- **D2S-VSE (Liu et al., 2025b):** A coarse-grained approach that leverages dense-to-sparse distillation with dense captions generated by a multimodal large language model (MLLM) to align cross-modal information capacity, and conducts retrieval via global embedding similarity.

**Image-text retrieval performance.** The quantitative results in Table 1 confirm that SEPS establishes new state-of-the-art performance across all evaluated settings. The core advantage lies in effectively bridging the semantic gap between modalities through dual-text semantic enhancement, particularly excelling in disambiguating complex visual scenes. On MS-COCO 5K with ViT-Base-224, SEPS achieves 16.4% and 29% improvements in bidirectional retrieval R@1 over LAPS, demonstrating that enriched textual semantics enable more discriminative cross-modal alignment than methods relying solely on sparse captions. Consistent gains across all backbones on Flickr30K further validate the scalability and robustness of our semantic enhancement mechanism.

**Compatibility with vision-language pre-training models.** To demonstrate the versatility of SEPS beyond task-specific training, we extend our framework to the widely-adopted CLIP model (Radford et al., 2021). As shown in Table 2, SEPS successfully enhances pre-trained vision-language representations, confirming that our dual-text semantic enhancement strategy is model-agnostic and complements existing pre-training paradigms. This validates that semantic enrichment addresses a fundamental challenge in cross-modal alignment rather than compensating for specific architectural limitations.

**Zero-shot transfer capability.** Table 3 presents zero-shot evaluations where models trained on Flickr30K are directly tested on MS-COCO without fine-tuning. SEPS substantially outperforms all fine-grained baselines, with 18.4% gain over LAPS in Image-to-Text R@1 on MS-COCO 1K, demonstrating that the semantic enhancement mechanism learns generalizable cross-modal representations rather than dataset-specific correlations. The superior zero-shot performance validates our hypothesis that enriching textual semantics fundamentally improves the model's understanding of visual-linguistic correspondence, enabling robust transfer across different data distributions and annotation styles.

Table 5: Comparison of different module ablations for SEPS framework on Flickr30K. We also show the results of the enhanced textual feature and relevance-aware alignment for our framework.

| Modules | Different Settings | Image-to-Text | | Text-to-Image | |
|---------|-------------------|:----:|:----:|:----:|:----:|
| | | R@1 | R@5 | R@1 | R@5 |
| SDTPS | only sparse text | 78.6 | 95.1 | 67.2 | 90.5 |
| | only dense text | 80.3 | 80.3 | 80.5 | 96.8 |
| | without aggregation | 85.2 | 96.1 | 84.7 | 97.3 |
| HRPA | only relevance-aware selection | 83.3 | 93.8 | 82.6 | 93.9 |
| | only mean value | 84.5 | 94.7 | 80.1 | 93.5 |
| | complete **SEPS** | **86.1** | **96.7** | **86.9** | **98.1** |

**Extension to visual grounding.** To verify whether enhanced semantic alignment benefits region-level understanding, we conduct zero-shot visual grounding experiments on RefCOCO, RefCOCO+, and RefCOCOg. As demonstrated in Table 4, SEPS consistently surpasses all baselines, with particularly notable gains on RefCOCO+ that excludes location-based expressions. This confirms that our framework captures semantic correspondence rather than exploiting spatial priors, evidencing that fine-grained semantic understanding acquired through dual-text integration effectively transfers to localization tasks without explicit spatial supervision. The results establish that semantic enhancement yields holistic improvements in cross-modal understanding beyond retrieval-specific optimizations.

## 4.5 ABLATION STUDY

We conduct comprehensive ablation studies on Flickr30K to systematically evaluate the contribution of each component. Results are presented in Table 5 and Figure 3.

**Semantic enhancement via dual-text integration.** The sparse-text-only baseline achieves 67.2% Text-to-Image R@1, while incorporating dense captions yields 80.5%, demonstrating a 13.3% improvement. The complete SDTPS module with aggregation further boosts performance to 86.9%, validating that unified semantic representations from complementary textual modalities effectively resolve visual ambiguities in complex scenes.

**Alignment strategy effectiveness.** The HRPA module combining relevance-aware selection with mean pooling achieves 86.9% Text-to-Image R@1, outperforming either strategy alone (82.6% and 80.1% respectively). This confirms that global semantic similarity and fine-grained correspondence detection are complementary mechanisms, where mean pooling captures holistic alignment while relevance-aware selection emphasizes critical patch-word pairs.

**Parameter sensitivity.** Figure 3 shows that SEPS maintains robust performance across different hyperparameter configurations, with moderate sensitivity to extreme patch selection ratios, particularly for ViT-based encoders.

## 5 CONCLUSION

In this work, we present the Semantic-Enhanced Patch Slimming (SEPS) framework, which systematically addresses patch redundancy and semantic ambiguity in fine-grained cross-modal alignment through strategic integration of MLLMs. The proposed SDTPS module employs a two-stage mechanism to fuse unified semantics from dense and sparse texts, while the HRPA module mitigates averaging bias through relevance-aware selection. Comprehensive experiments on Flickr30K and MS-COCO demonstrate that SEPS achieves state-of-the-art performance with 23%-86% improvements in rSum and strong zero-shot transfer capabilities. We acknowledge that our use of MLLM-generated dense text as semantic enrichment differs from traditional evaluation paradigms, though it follows established protocols and does not constitute classical data leakage. Additionally, dense text generation introduces preprocessing overhead, and MLLMs may perpetuate biases from training data. Future work should explore more efficient dense text generation methods, develop bias mitigation strategies, and extend our dual-text semantic enhancement paradigm to broader vision-language tasks such as visual question answering and image captioning, as well as integration with emerging foundation models to achieve both scale and precision in cross-modal understanding.

ETHICS STATEMENT

In this paper, we use Large Language Models to polish writing in Section 4 and appendix. The dense text used for supervision in our framework is generated by a large pre-trained MLLM. We acknowledge that these models may learn and perpetuate societal biases (e.g. gender and racial stereotypes) from their training data. Consequently, our method risks reinforcing these biases by relying on such models for visual guidance. We recognize this as a significant limitation and a key issue for future research to address.

REPRODUCIBILITY STATEMENT

To improve the reproducibility of our work, we upload our code, including all train logs, evaluate logs and best model checkpoints at https://anonymous.4open.science/r/SEPS/. The detailed settings for the hyperparameters are introduced in Section 4.3.

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

**Appendix to "SEPS: Semantic-enhanced Patch Slimming Framework for fine-grained cross-modal alignment"**

In this appendix, we provide the following materials:

A Comparison with Recent Competitive Vision-Language Pre-training Models;

B Comparison of image-text retrieval performance for SEPS with different hyperparameters on Flickr30K (referring to Section 4.4 and Section 4.5 in the main paper);

C Visualization of patch selection and alignment results.

# A  COMPARISON WITH RECENT COMPETITIVE VISION-LANGUAGE PRE-TRAINING MODELS

Table 6: Performance comparison on DCI Urbanek et al. (2024)and shareGPT4v Chen et al. (2023)datasets. All models use CLIP-ViT-Base-224 + CLIP-BERT-Base backbone.* means the zero-shot learning.

| Method | DCI | | shareGPT4v | |
|---|---|---|---|---|
| | Image-to-Text | Text-to-Image | Image-to-Text | Text-to-Image |
| CLIP (Radford et al., 2021)* | 45.5 | 43.0 | 78.2 | 79.6 |
| FineCLIP (Jing et al., 2024)* | 35.5 | 34.4 | 70.6 | 73.3 |
| FG-CLIP (Xie et al., 2025b)* | 61.8 | 60.6 | **96.7** | 94.9 |
| SigLIP 2 (Tschannen et al., 2025)* | 32.3 | 34.3 | 66.0 | 67.9 |
| FG-CLIP 2 (Xie et al., 2025a)* | **64.5** | 64.9 | 95.8 | 95.4 |
| **SEPS (ours)** | 62.4 | **65.1** | 96.0 | **96.5** |

Table 7: Performance comparison with large pre-training models on MS-COCO 5K and Flickr30k datasets using ViT-B Backbones.* means the zero shot learning.

| Method | MS-COCO 5K | | Flickr30k | |
|---|---|---|---|---|
| | Image-to-Text | Text-to-Image | Image-to-Text | Text-to-Image |
| CLIP (Radford et al., 2021)* | 51.8 | 32.7 | 82.2 | 62.1 |
| FineCLIP (Jing et al., 2024)* | 54.5 | 40.2 | 82.5 | 67.9 |
| FG-CLIP (Xie et al., 2025b)* | 64.1 | 45.4 | 90.7 | 76.4 |
| SigLIP 2 (Tschannen et al., 2025)* | 68.9 | 52.9 | 92.6 | 80.0 |
| FG-CLIP 2 (Xie et al., 2025a)* | 72.1 | 54.5 | **94.1** | 81.9 |
| **SEPS (ours)** | **73.9** | **73.5** | 86.1 | **86.9** |

To evaluate robustness, we benchmark SEPS against advanced pre-training models (e.g., FG-CLIP 2, SigLIP 2) across four datasets, with results presented in Table 6 and Table 7. We highlight a methodological distinction: baselines operate in a zero-shot regime pre-trained on billions of samples, whereas SEPS is fine-tuned on the 113k MS-COCO images. Despite this scale disparity, SEPS demonstrates exceptional efficacy. Specifically, Table 6 shows SEPS achieving 96.5% R@1 on ShareGPT4v, surpassing FG-CLIP 2. This performance advantage is most prominent on MS-COCO 5K, where SEPS attains 73.5% Text-to-Image R@1, significantly outperforming the zero-shot baseline (54.5%). These findings offer a nuanced perspective on the trade-off between scale and alignment granularity. While large-scale pre-training provides a powerful foundation for broad generalization, our results underscore that precise semantic alignment mechanisms remain pivotal for maximizing performance in downstream domains. SEPS effectively bridges the gap between general-purpose representations and task-specific precision. It demonstrates that by resolving patch-level ambiguity, researchers can unlock the full potential of standard backbones (e.g., ViT-B) with orders of magnitude less data. Thus, SEPS presents a resource-efficient paradigm for domain adaptation, allowing the academic community to achieve state-of-the-art results within accessible computational budgets.

# B    COMPARISON OF IMAGE-TEXT RETRIEVAL PERFORMANCE FOR SEPS WITH DIFFERENT HYPERPARAMETERS ON FLICKR30K.

Table 8: The comparisons of image-text retrieval for SEPS-Vit and SEPS-Swin with different selection ratio $\rho$ on Flicker30K.

| $\rho$ | Image-to-Text | | | Text-to-Image | | | rSum |
| | R@1 | R@5 | R@10 | R@1 | R@5 | R@10 | |
|---|---|---|---|---|---|---|---|
| *Vit-Base-224 + BERT-base, 14×14 patches* | | | | | | | |
| 0.1 | 85.6 | 91.9 | 97.0 | 85.1 | 97.0 | 98.6 | 553.4 |
| 0.2 | 86.5 | 92.4 | 96.6 | 86.3 | 97.5 | 98.9 | 557.3 |
| 0.3 | 87.4 | 92.9 | 96.2 | **87.5** | 98.0 | 99.2 | 561.2 |
| 0.4 | 87.0 | 94.2 | 97.0 | 87.1 | 98.3 | 99.3 | 562.8 |
| 0.5 | 86.5 | **95.4** | **97.8** | 86.6 | **98.6** | **99.4** | **564.3** |
| 0.6 | 87.0 | 94.1 | 97.3 | 86.9 | 98.3 | 99.2 | 562.8 |
| 0.7 | **87.5** | 92.8 | 96.8 | 87.3 | 97.9 | 99.0 | 561.3 |
| 0.8 | 86.9 | 92.6 | 96.4 | 86.8 | 97.9 | 99.1 | 559.8 |
| 0.9 | 86.2 | 92.5 | 96.1 | 86.3 | 97.8 | 99.2 | 558.2 |

To comprehensively evaluate our model's sensitivity and generality with respect to the key hyperparameter $\rho$, we conducted exhaustive experiments on both ViT and Swin backbones. The detailed results are presented in Table 8 and visually summarized in Figure 3. The analysis reveals both consistencies and notable distinctions in the performance trends across the two architectures. The ViT architecture exhibits a performance curve that is relatively sensitive to the value of $\rho$, where the rSum score reaches a distinct peak around $\rho = 0.5$ before declining at a comparatively rapid rate. In contrast, the Swin architecture demonstrates significant robustness; its performance curve maintains a near-peak level across a broad range of $\rho$ from 0.2 to 0.8, without showing sharp degradation. This comparative analysis shows that while moderate information slimming is beneficial for both backbones, our method possesses a very high tolerance for the choice of $\rho$ on the Swin architecture. Overall, this provides compelling evidence for the universality and high stability of our proposed method, highlighting its ability to readily adapt to different mainstream visual backbones without requiring meticulous hyperparameter tuning.

Table 9: The comparisons of image-text retrieval for SEPS with different settings of coefficients on Flicker30K. The best results are marked **bold**.

| $\lambda_1$ | $\lambda_2$ | Flickr30K 1K | | | | | | |
| | | Image-to-Text | | | Text-to-Image | | | rSum |
| | | R@1 | R@5 | R@10 | R@1 | R@5 | R@10 | |
|---|---|---|---|---|---|---|---|---|
| 0.25 | 0.25 | 85.8 | 94.3 | 97.8 | 83.1 | 97.6 | 98.7 | 557.4 |
| 0.25 | 0.5 | 85.0 | 92.9 | 96.6 | 85.0 | 97.5 | 98.7 | 555.7 |
| 0.25 | 0.75 | 84.7 | 92.4 | 96.2 | 85.6 | 97.7 | 98.9 | 555.5 |
| 0.25 | 1 | 84.8 | 86.9 | 94.0 | 83.2 | 97.0 | 98.7 | 544.6 |
| 0.5 | 0.25 | **86.8** | 96.7 | 97.9 | 85.5 | 97.9 | 99.1 | 563.9 |
| 0.5 | 0.5 | 86.5 | 95.4 | 97.8 | 86.6 | **98.6** | **99.4** | **564.3** |
| 0.5 | 0.75 | 85.6 | 91.2 | 95.7 | 86.1 | 97.7 | 99.0 | 555.3 |
| 0.5 | 1 | 86.6 | 92.5 | 96.9 | 86.3 | 98.3 | 99.3 | 559.9 |
| 1 | 0.25 | 85.3 | **96.9** | **98.9** | 69.7 | 93.0 | 97.0 | 540.9 |
| 1 | 0.5 | 86.4 | 94.7 | 98.1 | 86.7 | 98.3 | **99.4** | 563.7 |
| 1 | 0.75 | 86.0 | 94.5 | 97.1 | **87.3** | 98.2 | 99.3 | 562.4 |
| 1 | 1 | 86.1 | 93.7 | 96.9 | 86.9 | 98.1 | 99.2 | 560.9 |

We conducted a comprehensive sensitivity analysis on the loss coefficients $\lambda_1$ (sparse text) and $\lambda_2$ (dense text) to assess the robustness of our proposed SEPS framework. As demonstrated in Table 9, our model exhibits considerable stability across various coefficient combinations on the Flickr30K dataset. The overall performance metric rSum ranges from 540.9 to 564.3, representing a variation of approximately 4.2% relative to the optimal configuration, which underscores the model's

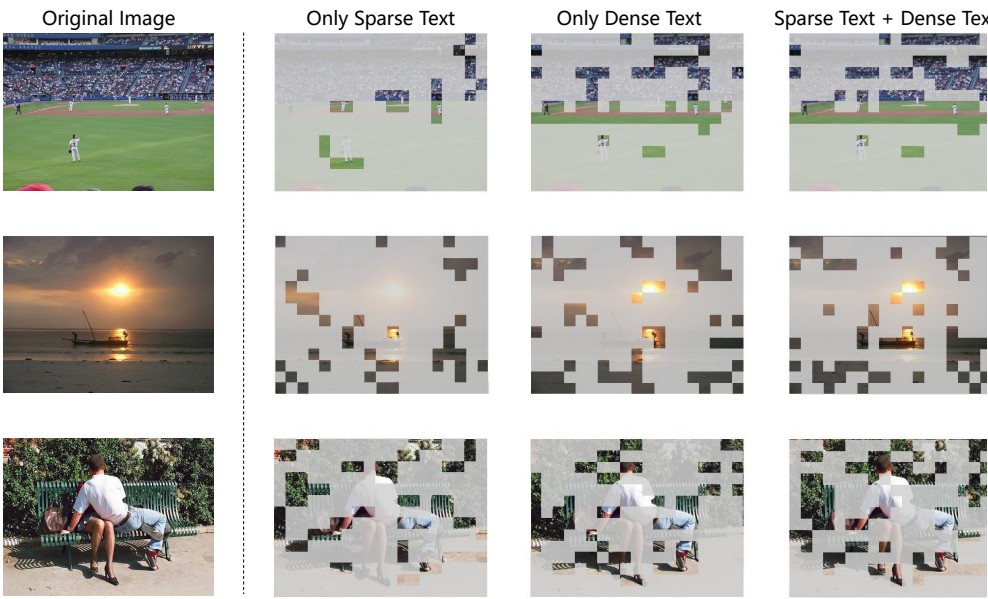

Figure 4: The visualization of visual patch selection with different combinations of sparse text and dense text.

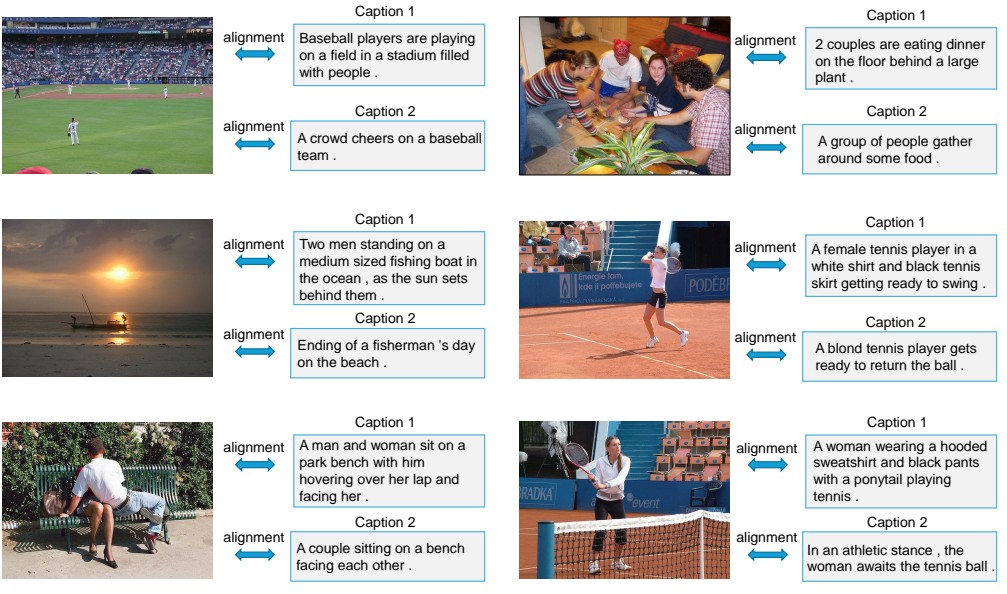

Figure 5: The visualization of cross-modal alignment results of SEPS.

inherent robustness to hyperparameter selection. Notably, the optimal configuration ($\lambda_1 = 0.5$, $\lambda_2 = 0.5$) achieved the highest rSum of 564.3, demonstrating balanced performance across both retrieval directions. This configuration also yielded superior results in text-to-image retrieval, attaining R@5 and R@10 scores of 98.6% and 99.4%, respectively. Conversely, for image-to-text retrieval, the setting ($\lambda_1 = 0.5$, $\lambda_2 = 0.25$) achieved the highest R@1 performance of 86.8%, while the ($\lambda_1 = 1$, $\lambda_2 = 0.25$) configuration excelled in R@5 and R@10 metrics with scores of 96.9% and 98.9%, respectively. In the challenging text-to-image R@1 task, the ($\lambda_1 = 1$, $\lambda_2 = 0.75$) configuration delivered the peak performance of 87.3%.

These results reveal that while individual metrics may favor specific coefficient combinations, the overall model maintains consistently high performance across the parameter space. This stability pattern strongly indicates that the superior performance of SEPS stems from its fundamental architectural design principles rather than from aggressive hyperparameter optimization, thus validating the effectiveness of our semantic guidance mechanism utilizing dense textual representations.

## C  VISUALIZATION

To intuitively illustrate the internal mechanism and final efficacy of our SEPS framework, we provide a qualitative visual analysis. Figure 4 clearly reveals the visual patch selection process. When guided solely by sparse text, the model identifies primary objects, but the selection can be coarse. Conversely, dense text alone can add detail but may sometimes overemphasize secondary regions. In contrast, our SEPS framework achieves a more precise selection of visual evidence by being the first to combine MLLM-generated dense text with original sparse captions. This combination allows the model to effectively bridge the information density gap between modalities, fusing the global context from sparse captions with the granular detail from dense descriptions to accurately preserve all semantically relevant patches. This superior patch selection capability translates directly into more robust fine-grained alignment, as demonstrated in Figure 5. Our model successfully aligns a single complex image with multiple, semantically diverse, yet correct captions, such as understanding both "Baseball players are playing" and "A crowd cheers on a baseball team." In summary, these visualizations intuitively demonstrate our core insight. By systematically leveraging dense text to assist visual patch selection, the SEPS framework achieves a more comprehensive scene understanding, which is the key to its state-of-the-art recall in complex fine-grained alignment tasks.

