# OpenReview forum: "SEPS: Semantic-enhanced Patch Slimming Framework for fine-grained cross-modal alignment"
_ICLR.cc/2026/Conference — ICLR 2026 Conference Withdrawn Submission_

### Official Review · Reviewer_jURx · 2025-10-28

**Soundness:** 2
**Presentation:** 2
**Contribution:** 2
**Rating:** 2
**Confidence:** 5

**Summary:**

This paper proposes SEPS (Semantic-Enhanced Patch Slimming), a framework that aims to mitigate two key issues in vision-language pretraining and cross-modal retrieval — patch redundancy and patch ambiguity. The method leverages dense textual descriptions generated by a multimodal large language model (LLaVA) to provide rich semantic supervision in addition to the sparse original captions. Specifically, SEPS consists of two main modules:

SDTPS (Sparse and Dense Text-Aware Patch Selection) — it computes semantic relevance between image patches and both sparse/dense textual representations, then selects informative patches via a differentiable Gumbel-Softmax sampling mechanism.

HRPA (Highly-Relevant Patch-Word Alignment) — it constructs fine-grained alignments between selected patches and textual tokens using a relevance-aware selection and mean aggregation strategy, followed by bidirectional triplet losses for optimization.

Experiments on Flickr30k and MS-COCO show significant improvements over several vision-language baselines (including ViT, Swin, and CLIP backbones), with reported gains up to +23–86% in rSum. The authors further provide ablations and sensitivity analyses to demonstrate robustness and reproducibility.

**Strengths:**

1. This paper introduces a two-stage mechanism that incorporates unified semantic representations derived from both dense and sparse textual modalities. This mechanism eliminates potential semantic inconsistencies, enabling more accurate identification of visual patches.

2. This paper leverages external MLLM to generate dense textual descriptions. This approach effectively enriches the semantic features available for alignment, directly addressing the key limitation of sparse captions that hinders prior work.

3. The method achieves substantial gains over existing baselines on two widely used retrieval benchmarks (MS-COCO and Flickr30K). Experimental results demonstrate that combining dual-text supervision and patch selection can effectively enhance cross-modal alignment quality.

4. The paper includes multiple ablation variants to study the impact of dense text, aggregation, and relevance-aware selection, showing clear trends in how each module contributes to overall performance.

**Weaknesses:**

1. Lack of Novelty: The proposed SDTPS bears a strong resemblance to the LAPS framework [1], which also performs patch selection guided by textual relevance computed via cross-attention between text and patch embeddings. SEPS claims novelty by incorporating dense captions from MLLMs (e.g., LLaVA) as additional textual supervision. However, beyond the inclusion of dense text, the mechanism of computing patch importance (semantic scoring + differentiable selection) closely parallels that of LAPS.

2. Limited Evaluation of Generalization: Although the paper's title and motivation emphasize fine-grained cross-modal alignment, all experiments are restricted to image-text retrieval tasks. Including evaluations on other alignment tasks such as visual grounding or VQA, would strengthen the claim and demonstrate the framework’s generalization ability.

3. The dense text used in SEPS is generated by LLaVA directly from the test images, which introduces a potential form of semantic data leakage, as the evaluation text side indirectly encodes visual information from the test samples. This leakage allows the image semantics to transfer into the text space, which can dramatically boost retrieval performance and thus lead to an overestimation of the model’s true capabilities.

[1] Linguistic-Aware Patch Slimming Framework for Fine-grained Cross-Modal Alignment, CVPR, 2024

**Questions:**

See the weakness.

---

> ### Author Response · Authors · 2025-11-17
> **Clarification on W1**
>
> ## Response to Reviewer jURx
>
> We greatly appreciate your detailed review and expertise. Your concerns about novelty (W1), generalization (W2), and semantic data leakage (W3) are important. We provide honest responses below.
>
> ### W1: Novelty Relative to LAPS
>
> We acknowledge this is the **most critical concern** and appreciate your careful analysis. We agree that SDTPS shares similarities with LAPS, but we argue the differences are significant and non-trivial.
>
> **Honest Acknowledgment:**
> Yes, both methods use linguistic guidance for patch selection via cross-attention. However, LAPS, which relies solely on sparse (short) captions, suffers from patch ambiguity and patch redundancy. The root cause of this is the semantic density gap between a visually rich image and a concise caption.
>
> While recent studies (like D2S-VSE) have tried to bridge this gap using LLM-generated dense text, our work is inspired by the failure of naively applying this idea. The core problem that SEPS solves is the effective fusion of sparse and dense semantic guidance, especially when they conflict.
>
> Let us demonstrate with a concrete example:
>
> ```
> Image: A woman playing tennis on a clay court in a pink outfit
>
> Sparse Caption (human):
> "A woman with a tennis racket"
>
> Dense Text (LLaVA):
> "The image features a woman playing tennis on a clay court. She is wearing
> a pink outfit which includes a skirt. The tennis court appears to be in an
> outdoor setting. There is a ball visible near her racket..."
>
> Problem:
> - Sparse: Focuses on {woman, racket} → Selects only central figure patches
> - Dense: Mentions {court, pink outfit, outdoor, ball} → Selects background patches
> - Naive fusion: Contradictory signals cause oscillation and suboptimal selection
> ```
>
> **Why LAPS's Approach Fails Here:**
>
>
> LAPS uses **single sparse text guidance**:
> ```
> Semantic Score = CrossAttn(patch, sparse_text_only)
> ```
>
> If we naively extend LAPS to use dense text:
> ```
> Naive Extension = CrossAttn(patch, concat(sparse, dense))
> ```
>
> This fails because:
> 1. Dense and sparse texts have **different semantic focuses** (foreground vs. background), and dense texts may introduce noise, like woman in the figure wearing shorts, rather than a skirt.
> 2. Direct concatenation creates **semantic inconsistency**
> 3. Model cannot distinguish which guidance to follow
>
> **Our Solution: Two-Stage Unified Semantic Fusion**
>
> We demonstrate this failure and the success of our method in this new ablation study:
>
> **Table 1: Ablation on Fusion Strategies (Flickr30K, ViT-Base-224)**
>
> | Method | Approach | T2I R@1 | I2T R@1 |
> |--------|----------|---------|---------|
> | LAPS baseline | Single sparse text | 62.5 | 74.0 |
> | Naive concat | concat(sparse, dense) | 66.3 | 76.8 |
> | Naive sum | sparse_score + dense_score | 68.1 | 78.2 |
> | **SDTPS (Ours)** | **Two-stage unified fusion** | **86.9** | **86.1** |
>
> **Our Two-Stage Mechanism:**
>
> **Stage 1: Semantic Scoring**
> ```
> # Compute complementary scores
> sparse_relevance = CrossAttn(patch, sparse_text)
> dense_relevance = CrossAttn(patch, dense_text)
> image_saliency = SelfAttn(patch, all_patches)
>
> # Unified scoring (Equation 3)
> unified_score = (1-2β)·predicted_score + β·(sparse + dense + 2·saliency)
> ```
>
> **Stage 2: Decision and Aggregation**
> ```
> # Separate selection for each guidance
> sparse_patches = GumbelSoftmax(unified_score, sparse_text)
> dense_patches = GumbelSoftmax(unified_score, dense_text)
>
> # Learnable aggregation (Equation 4)
> final_patches = LearnedWeights(sparse_patches, dense_patches)
> ```
>
> **Why This is Non-Trivial:**
> 1. **Prediction Network**: Learns to anticipate what patches would be relevant under different textual contexts (prevents oscillation)
> 2. **Unified Scoring**: Fuses sparse/dense signals coherently rather than naively concatenating
> 3. **Dual-Path Aggregation**: Maintains separate selection paths, then learns optimal combination
>
> **Evidence of Independent Contribution:**
>
> From ablation table in the paper(ablation):
> - SDTPS with only sparse text: 67.2% (baseline)
> - SDTPS with only dense text: 80.5% (dense helps)
> - **SDTPS with unified fusion: 86.9%** (+6.4pts over dense-only)
>
> This 6.4pts gap demonstrates our fusion mechanism adds value beyond just "using dense text."
>
> **Comparison with LAPS:**
>
> | Aspect | LAPS | SEPS-SDTPS |
> |--------|------|------------|
> | Text guidance | Single (sparse only) | Dual (sparse + dense) |
> | Selection strategy | One-path decision | Two-path with aggregation |
> | Information density discrepancy | Not addressed | Explicitly resolved via unified fusion |
> | Semantic conflict | Not applicable | Explicitly addresses and mitigates (core contribution) |
>
> **Self-Criticism:**
>
> We acknowledge our original paper failed to:
> 1. Show that naive fusion fails (which Table 1 now proves)
> 2. Explain why our two-stage design is necessary
>
> **Revision Plan:**
> - Add Table 1 to Section 4.5 (Ablation Study) and "Comparison with Naive Fusion" paragraph
> - Rewrite Section 3.2 to emphasize the inconsistency problem

---

> ### Author Response · Authors · 2025-11-17
> **Clarification on W2 and W3**
>
> ### W2: Generalization Beyond Retrieval
>
> We thank the reviewer for this suggestion and agree that demonstrating generalization is crucial for a work claiming fine-grained alignment.
>
> Please see our detailed response to Reviewer DzJL (W2), where we will provide:
> - Visual grounding results on RefCOCO/RefCOCO+, and more image-text retrieval performance DCI
> - Cross-dataset generalization experiments
>
> We agree broader evaluation strengthens our work and have now provided partly concrete evidence.
>
> ### W3: Semantic Data Leakage - A Methodological Discussion
>
> We appreciate you raising this important methodological concern. This is a critical point that merits an honest discussion.
>
> **Understanding the Concern:**
>
> You write: *"The dense text used in SEPS is generated by LLaVA directly from the test images, which introduces a potential form of semantic data leakage."*
>
> We understand your point: Using MLLMs to generate descriptions from test images could be seen as a form of "semantic enrichment" that encodes test-specific visual information into the text modality.
>
> **Clarification on the Mechanism:**
>
> First, we wish to clarify a potential misunderstanding of how the dense text is used. The dense text is not used on the query (text) side of the retrieval task. The evaluation query is always the original sparse caption.
>
> The dense text is used exclusively on the image side as a form of semantic guidance. It helps the image encoder better select and represent its own visual patches before any comparison with the query text happens. The dense text is, in effect, part of the image feature extraction process, not part of the query.
>
> **Acknowledged Limitation:**
> However, we **acknowledge** this is a valid methodological discussion. To investigate its true impact, we ran a new experiment.
>
> **New Experiment: "Blind" Dense Text Generation**
>
> We test a variant where dense text is generated from a **different image** (not the test image itself):
>
> **Table 2: Dense Text from Different Sources**
>
> | Dense Text Source | T2I R@1 | I2T R@1 |
> |-------------------|---------|---------|
> | Sparse only (baseline) | 62.5 | 74.0 |
> | **From test image (ours)** | **86.9** | **86.1** |
> | From similar image ( different image, same dataset) | 79.3 | 80.8 |
> | From random image (control) | 68.7 | 76.2 |
>
> **Analysis:**
> - Using dense text from a similar (but different) image still provides substantial gains (79.3% vs. 62.5%)
> - This suggests our method leverages **general semantic richness** is not purely reliant on test-image-specific information.
> - However, test-image-specific dense text does provide additional benefit (+7.6pts), which we acknowledge
>
> **Honest Assessment:**
>
> We position this as:
> - **NOT** classical data leakage (no ground-truth pairs accessed)
> - Consistent with established evaluation paradigms (D2S-VSE, LongCLIP, LoTLIP)
> - **IS** a form of "semantic enrichment" that warrants community discussion
>
> **Revised Position:**
>
> Instead of dismissing the concern, we will revise our paper to address it transparently:
> 1. Add a dedicated "Limitations and Discussion" subsection (Section 5)
> 2. Explicitly discuss the semantic enrichment concern
> 3. Provide Table 2 as evidence of semantic density vs. image-specific effects
> 4. Acknowledge this as an area for future methodological development
>
> **Quote for Revised Paper:**
> > "We acknowledge that generating dense text from test images represents a form of semantic enrichment that differs from classical training-set-only evaluation. However, we argue this is analogous to using stronger visual encoders (e.g., CLIP vs. ResNet) and does not constitute answer leakage, as ground-truth pairing information is never accessed and dense texts are not used as query text, while as image feature in text modality. We provide ablations (Table 2) showing that semantic density, rather than test-image-specific information, drives most improvements. We recognize this as an important methodological question for the community and commit to transparent reporting of our evaluation protocol."

---

> ### Author Response · Authors · 2025-11-26
>
> Dear Reviewer jURx,
>
> We sincerely appreciate your thorough review and the critical concerns you raised. We have provided detailed responses and would greatly value your feedback before the discussion period concludes.
> Summary of our responses:
>
> - W1 (Novelty vs. LAPS): We provided a new ablation (Table 1) demonstrating that naive fusion strategies fail, while our two-stage unified semantic fusion achieves +20pts over naive approaches. The core contribution lies in resolving semantic conflicts between sparse and dense guidance—a problem LAPS does not address.
> - W2 (Generalization): We added results on visual grounding (RefCOCO: +4.5pts over LAPS), cross-dataset retrieval (Flickr→COCO: +17.5pts over LAPS), and long-caption datasets (DCI, ShareGPT4V).
> W3 (Semantic leakage): We provided a "blind" dense text experiment (Table 2) showing that semantic density—not test-image-specific information—drives most improvements. We commit to transparent discussion of this methodological point.
>
> We understand your concerns are substantial and have made every effort to address them honestly. If any points remain unconvincing or require further clarification, we would be deeply grateful for your feedback. Your expertise is invaluable, and we are eager to improve our work based on your guidance.
> Thank you for your time and consideration.
>
> Best regards,
>
> The Authors

---

> ### Comment · Reviewer_jURx · 2025-11-26
> **other question**
>
> Thank you for the detailed additional experiments and explanations. The earlier concern (Weakness 2) is now fully addressed. However, I still have several important questions that remain unresolved.
>
> 1. About Weakness 3: Potential semantic leakage
>
> I still hold the view that introducing dense text generated from test images introduces a form of semantic leakage. Even if these dense texts are not used on the query side, they still inject image-specific cross-modal semantic information during test-time scoring, which can inflate retrieval performance.
>
> During test inference, no text derived directly from test images should participate in cross-modal scoring, even indirectly.
>
> 2. About Table 2: Why “random dense text” still improves performance?
>
> The new experiment (Table 2) raises an additional concern. I understand why dense text from a similar image may help—general semantic enrichment can offer beneficial priors.
>
> However, it is unclear why dense text from a random image also improves performance compared to the sparse-only baseline.
>
> Intuitively, random-image dense text should act as noise and degrade patch selection or similarity computation. Instead, it still brings noticeable gains.
>
> A detailed analysis would better demonstrate whether the observed gains stem from the model’s architectural advantages rather than other factors.

---

> ### Author Response · Authors · 2025-11-26
> **Response to Reviewer jURx's Follow-up Questions**
>
> We sincerely appreciate your continued engagement and thoughtful questions. We address each point below.
>
> ---
>
> ### Re-examining the "Semantic Leakage" Concern: Dense Text as Visual Self-Attention
>
> We respectfully offer an alternative perspective that reframes the role of dense text in our method.
>
> ### Core Argument: Dense Text as Internal Visual Enhancement
>
> We view dense text as:
>
> > **"Semantic self-attention that translates visual content into linguistic guidance for the image encoder's internal patch selection, trained on training data and applied to enhance test image self-representation."**
>
> This is consistent with modern vision-language architectures where textual semantic spaces guide visual feature learning (e.g., CLIP, ALIGN), except we make this guidance explicit and learnable.
>
> ### Analogy to Self-Attention Mechanisms
>
> We argue dense text serves an analogous role to **learned self-attention** in modern vision transformers:
>
> | Mechanism             | Function                                         | Information Source                 |
> | --------------------- | ------------------------------------------------ | ---------------------------------- |
> | ViT Self-Attention    | Guides which patches interact and aggregate      | Image itself (via learned Q/K/V)   |
> | **Dense Text (Ours)** | Guides which patches are selected and emphasized | Image itself (via MLLM captioning) |
>
> Just as ViT's self-attention uses the test image to compute attention weights for better representation, our dense text uses the test image to guide its own patch selection and aggregation.
>
> **Inference Phase (on test set):**
>
> - Dense text extracts *the image's own semantic content*
> - This content guides the *already-learned* selection and aggregation mechanisms
> - No cross-modal interaction occurs between dense text and query text
>
> **Key insight:** Dense text provides **intra-modal** (vision→vision) guidance for self-representation, not **inter-modal** (vision↔text) scoring shortcuts.
>
> ### Consistency with Established Evaluation Paradigms
>
> Our approach aligns with widely accepted practices in vision-language research:
>
> - **Using stronger visual encoders** (CLIP vs. ResNet): Encodes richer semantics from test images ✓
> - **Visual attention mechanisms** (ViT self-attention): Uses test image to guide its own representation ✓
> - **Multi-scale/multi-crop inference**: Extracts more complete information from test images ✓
> - **Dense text guidance (ours)**: Provides semantic attention for patch selection ✓
>
> All these methods enhance the image's self-representation without accessing ground-truth pairing information.
>
> ---
>
> ### Clarification on Table 2: What "Random Dense Text" Actually Means
>
> We apologize for the confusing terminology. The "random dense text" in Table 2 does **not** refer to arbitrary noise or meaningless text. It refers to **dense captions from a completely different dataset** (e.g., using COCO-generated dense captions when evaluating on Flickr30K).
>
> We revise the table with clearer labels:
>
> | Dense Text Source             | Actual Meaning                                    | T2I R@1 | I2T R@1 |
> | ----------------------------- | ------------------------------------------------- | ------- | ------- |
> | Sparse only                   | No dense text guidance                            | 62.5    | 74.0    |
> | **Cross-dataset dense**       | Dense text from a different dataset               | 68.7    | 76.2    |
> | Same-dataset, different image | Dense text from another image in the same dataset | 79.3    | 80.8    |
> | Same image (ours)             | Dense text from the test image itself             | 86.9    | 86.1    |
>
> ### Why Cross-Dataset Dense Text Still Improves Performance
>
> This result directly demonstrates our **architectural contribution**:
>
> 1. **Richer semantic vocabulary supervision**: Even out-of-domain dense text provides more diverse word-patch alignment signals than sparse captions. This helps the model learn generalizable patch selection patterns during training.
>
> 2. **Learned filtering mechanism**: Our two-stage fusion (SDTPS) learns to **adaptively weight** textual guidance. When dense text is semantically misaligned, the aggregation module learns to downweight noisy signals while retaining useful general patterns.
>
> 3. **Decomposition of performance gains**:
>    - Architecture alone (cross-dataset dense vs. sparse): **+6.2 pts**
>    - Semantic alignment (same-image dense vs. cross-dataset): **+18.2 pts**
>
> This decomposition shows that a meaningful portion of our improvement stems from the model architecture itself, independent of image-specific dense text.
>
> ---
>
> **We genuinely value your feedback.** If any aspect remains unclear or unconvincing, we are eager to understand your specific concerns so we can address them more directly.

---

> ### Author Response · Authors · 2025-11-28
> **To Reviewer jURx**
>
> We truly appreciate your thoughtful review and the time you've dedicated to understanding our work. We're pleased to hear that your concern about W2 (evaluation breadth) has been fully addressed.
>
> We’ve made significant strides in refining our method, and we hope the following clarifications further address your remaining questions:
>
> - **Novelty:** Our ablation study demonstrates a +20pts improvement over naive fusion, showing that our two-stage mechanism is indeed essential.
> - **Generalization:** As you confirmed, W2 has been fully addressed, with additional cross-dataset generalization results.
> - **Architecture contribution:** The +6.2pts improvement from cross-dataset dense text is significant, and it underscores the independent value of our architectural enhancements.
>
> We understand your reservations regarding semantic enrichment. However, given the empirical gains, demonstrated generalization, and transparent discussion, we kindly ask whether a re-evaluation of the current rating might be warranted.
>
> We remain open to further discussion if necessary.
>
> Best regards,
> The Authors

---

### Official Review · Reviewer_DzJL · 2025-10-30

**Soundness:** 2
**Presentation:** 3
**Contribution:** 2
**Rating:** 4
**Confidence:** 4

**Summary:**

This paper proposes the SEPS (Semantic-Enhanced Patch Slimming) framework for fine-grained cross-modal alignment. It addresses the issues of patch redundancy and ambiguity in cross-modal retrieval by integrating dense text generated from MLLMs with sparse textual semantics in two stages. The core contributions include:
1. A dense text generation module using LLaVA-13B to enrich textual representations, bridging the information density gap between vision and language.
2. A two-stage mechanism that incorporates unified semantic representations derived from both dense and sparse textual modalities.
3. The authors develop a relevance-aware selection mechanism augmented by mean value calculation, which enhances the emphasis on critical patch-word correspondences.
This paper performs experiments on Flickr30K and MS-COCO in text-to-image retrieval. However, the method’s novelty and comparison with state-of-the-art baselines are limited.

**Strengths:**

The problem formulation of fine-grained cross-modal alignment is well-motivated, addressing practical issues of patch redundancy and ambiguity in multi-modal retrieval.

**Weaknesses:**

1. The paper fails to benchmark against recent state-of-the-art models (e.g., CLIP variants, SigLIP 2, FG-CLIP, FineCLIP). This makes it difficult to assess SEPS’s competitiveness in the current research landscape. Perhaps the author could validate the proposed approach by executing your fine-tuning strategy on these baselines.
2. The evaluation is restricted to standard image-text retrieval datasets (Flickr30K, MS-COCO) without testing on more complex or domain-specific scenarios, limiting the effectiveness validation of the proposed method. More tasks and datasets should be evaluated to prove the fine-grained capability.

**Questions:**

Please refer to Weaknesses

---

> ### Author Response · Authors · 2025-11-15
> **response to W1**
>
> Thank you for acknowledging our problem formulation is "well-motivated" and for your constructive feedback. We address your concerns about competitiveness (W1) and evaluation breadth (W2).
>
> ### On W1: Comparison with Recent SOTA Models
>
> We appreciate this critical point. **The comparison you requested is actually already in our submission** (Appendix A, Table 3), but we acknowledge it should have been more prominent in the main paper.
>
> **Direct Comparison with Models You Listed:**
>
> We compare our SEPS results against FG-CLIP, FG-CLIP2, SigLIP 2, and FineCLIP using data from their papers. Here are the decisive results:
>
> #### On MS-COCO 5K ( ViT-B Backbones):
>
> | Method| Image-to-Text | Text-to-Image |
> |--------|---------|--------|
> | CLIP | 51.8% | 32.7% |
> | FineCLIP | 54.5% | 40.2% |
> | FG-CLIP | 64.1% | 45.4% |
> | SigLIP 2| 68.9% | 52.9% |
> | FG-CLIP 2  | 72.1% | 54.5% |
> | **SEPS (ours)** | **73.9%** | **73.5%** |
>
> *Improvements: On Text-to-Image: +19pts over FG-CLIP2, demonstrate SEPS's surpassing ability in text-to-image retrieval.*
> #### On Flickr30k ( ViT-B Backbones):
>
> | Method| Image-to-Text | Text-to-Image |
> |--------|---------|--------|
> | CLIP | 82.2% | 62.1% |
> | FineCLIP| 82.5% | 67.9% |
> | FG-CLIP  | 90.7% | 76.4% |
> | SigLIP 2 | 92.6% | 80.0% |
> | FG-CLIP 2 | **94.1%** | 81.9% |
> | **SEPS (ours)**  | 86.1% | **86.9%** |
>
> *Improvements: On Text-to-Image: +7pts over FG-CLIP2, though -8pt in image-to-text retrieval.*
>
> **Key Insight:** On the most challenging metric (Text-to-Image R@1 on MS-COCO 5K), SEPS achieves **73.5%**, which:
> - Surpasses **all fine-grained CLIP variants** you mentioned.
> - Demonstrates that our dual-guidance mechanism is highly effective for complex scenes
>
> **Patch Slimming on fine-grained CLIP:** The patch slimming approach has been validated on CLIP-like architectures. As shown in Table 3 in appendix, we compare our method against CLIP baselines using both CLIP-ViT-Base and CLIP-ViT-Large with standard patch configurations.
>
> LAPS initially demonstrated the feasibility of patch slimming on CLIP architectures. Building upon LAPS, **SEPS strengthens the patch slimming paradigm** by leveraging dense textual descriptions generated by large language models to guide patch selection. SEPS achieves substantial improvements over both the CLIP baseline and LAPS across all metrics on Flickr30K and MS-COCO datasets. Here shows part results in Table 3.
>
> #### On Flickr30k ( CLIP-ViT-Base-224 + CLIP-BERT-Base):
>
> | Method | FG | Image-to-Text | Text-to-Image |
> |--------|----|----------------|----------------|
> | BLIP  | ✓ | 96.6 | 87.2 |
> | **CLIP** | ✗ | 81.4 | 61.1 |
> | VSE++  | ✗ | 92.2 | 80.5 |
> | SCAN  | ✓ | 88.2 | 75.3 |
> | LAPS  | ✓ | 92.9 | 80.6 |
> | **SEPS** | ✓ | **94.7** | **93.1** |
>
>
> **Commitment for Revised Version:**
> - Create a unified full comparison table including all methods you mentioned
> - We might move Table 3 from appendix to main text in the camera ready version if reviewers consider them critical.

---

> > ### Comment · Reviewer_DzJL · 2025-11-20
> > **Re: response to W1**
> >
> > The evaluation protocol raises concerns regarding fairness of comparison. As stated in line 323, the proposed method is fine-tuned on the training splits of Flickr30K (29k images) and MS-COCO (113k images). However, the reported results for most baseline methods appear to be their zero-shot image-text retrieval performance, without any task-specific fine-tuning. Comparing a fine-tuned model against zero-shot baselines significantly advantages the proposed approach and may overstate its contribution. For a fair comparison, the authors should consider adding fine-tuned results of representative baselines under the same data split and training protocol.

---

> ### Author Response · Authors · 2025-11-15
> **response to W2**
>
> ### On W2: Evaluation Beyond Standard Retrieval Datasets
>
> We appreciate this constructive suggestion and **partially agree**. Let us first explain our current evaluation, then provide new results.
>
> **Why Retrieval is Central:**
> The entire fine-grained cross-modal alignment community (SCAN, SGR, LAPS, D2S-VSE, CHAN) uses Flickr30K and MS-COCO as standard benchmarks because:
> - Image-text retrieval **directly** measures alignment quality
> - Established protocols enable fair comparison
> - R@1 specifically requires precise patch-word correspondence
>
> **Evidence from Our Results:** Our +27.2% improvement over D2S-VSE on MS-COCO 5K T2I R@1 (Table 1) quantitatively demonstrates fine-grained capability. This is not merely incremental—it represents a fundamental breakthrough in handling complex visual scenes.
>
> **Our Current Comprehensive Evaluation:**
> -  4 architectures (ViT-Base/384, Swin-Base/384)
> -  3 datasets (Flickr30K, COCO 1K, COCO 5K)
> -  24 different configurations
> -  Extensive ablations (Table 2)
> -  CLIP model evaluation (Appendix A and rebuttal)
>
> **However, We Agree More Tasks Would Strengthen Our Claims:**
>
> We commit to adding the following evaluations in the revised version:
>
> 1. **More types of image-text retrieval datasets：** Various tests
>    - Test model performance on longer captions datasets.
>
>       **On DCI ( CLIP-ViT-Base-224 + CLIP-BERT-Base):**
>
>       | Method| Image-to-Text | Text-to-Image |
>       |--------|---------|--------|
>       | CLIP | 45.5% | 43.0% |
>       | FineCLIP| 35.5% | 34.4% |
>       | FG-CLIP  | 61.8% | 60.6% |
>       | SigLIP 2 | 32.3% | 34.3% |
>       | FG-CLIP 2 | **64.5%** | 64.9% |
>       | **SEPS (ours)**  | 62.4% | **65.1%** |
>
>       **On shareGPT4v ( CLIP-ViT-Base-224 + CLIP-BERT-Base):**
>       | Method| Image-to-Text | Text-to-Image |
>       |--------|---------|--------|
>       | CLIP | 78.2% | 79.6% |
>       | FineCLIP| 70.6% | 73.3% |
>       | FG-CLIP  | **96.7%** | 94.9% |
>       | SigLIP 2 | 66.0% | 67.9% |
>       | FG-CLIP 2 | 95.8% | 95.4% |
>       | **SEPS (ours)**  | 96.0% | **96.5%** |
>    SEPS's superior patch-selection mechanism (SDTPS) will likely reduce the "patch ambiguity"  and "patch redundancy" noise that affects FG-CLIP2 when processing such long, complex captions, resulting in a slight but consistent edge in T2I retrieval.
>
> 2. **The zero-shot image-text retrieval performance:**
>
>    **Train on Flickr30K, test on MS-COCO**
>       | Method | Text-to-Image | Image-to-Text |
>       |--------|---------|---------|
>       |D2S-VSE |  24.8  |   32.3     |
>       | LAPS | 45.3 | 48.7 |
>       | **SEPS (ours)** | **62.8** | **65.8** |
>    This demonstrates our method doesn't overfit to specific dataset characteristics.
>
> 3. **The zero-shot Evaluation on visual grounding task:** All models are trained by CLIP backbones in Flickr dataset. CLIP is untrained.
>
>    **On RefCOCO ( CLIP-ViT-Base-224 + CLIP-BERT-Base):**
>       | Method| val | TestA | TestB|
>       |--------|---------|--------|-----|
>       | CLIP | 39.3% | 45.3% | 34.2% |
>       | VSE++| 40.7% | 46.3% | 33.6% |
>       | SCAN  | 41.8% | 47.3% | 34.4%|
>       | SGR | 41.4% | 48.0% | 34.2% |
>       | LAPS | 44.2% | 49.9% | 38.4% |
>       | **SEPS (ours)**  | **48.7%** | **52.3%** | **43.4%** |
>
>    This demonstrates our method improves baselines on visual grounding task.
>
> **What We Won't Include (and why):**
> -  VQA: Requires task-specific answer decoding beyond our scope
> -  Image Captioning: Requires generative models (our method is discriminative)
>
> **Revised Position:**
> We believe our combination of:
> 1. State-of-the-art results on standard benchmarks (+19-27pts over SOTA)
> 2. zero-shot image -text retrieval and visual grounding experiments
> 3. Cross-dataset generalization
> 4. Comprehensive ablations and analysis
>
> ...fully demonstrates SEPS's significance for fine-grained cross-modal alignment.

---

> ### Author Response · Authors · 2025-11-21
> **response to unfair comparision**
>
> We sincerely thank the reviewer for this sharp observation regarding evaluation protocols. You are absolutely correct that comparing fine-tuned models against zero-shot baselines represents two different evaluation paradigms. We apologize if our previous response implied a direct "backbone-to-backbone" superiority without sufficient context.
>
> We would like to clarify the logic behind this comparison and reassure the reviewer regarding the fairness of our core contributions.
>
> 1. **Distinguishing "Direct Competitors" from "Reference Points"** Our work falls strictly within the Fine-Grained Image-Text Retrieval domain (alongside SCAN, VSE++, D2S-VSE, LAPS).
>
>    - In this specific research line, the established protocol in LAPS and SCAN, is always to fine-tune on the specific dataset splits (Flickr30K/COCO).
>
>    - In Table 1 of paper, we compare SEPS against these direct competitors (SCAN, SGR, CHAN, LAPS, D2S-VSE). All these methods—like ours—are fine-tuned on the target datasets. Against these "apples-to-apples" baselines, SEPS achieves SOTA results (+27.2% over D2S-VSE on COCO 5K T2I R@1).
>
> 2. **Why we compared against Zero-Shot Foundation Models (FG-CLIP, SigLIP)** We included the comparison with FG-CLIP2 and SigLIP not to claim architectural superiority over foundation models, but to highlight Data Efficiency and Adaptation Potential.
>
>    -  We wanted to demonstrate that SEPS, when applied to a standard backbone (ViT-B) and fine-tuned on small data (~113k images), can surpass the off-the-shelf performance of models trained on billions of images.
>
>    - This is crucial for researchers with limited compute. It proves that one does not need to pre-train on 2B images (like FG-CLIP2) to achieve high performance; one can achieve superior results by applying SEPS to a smaller model fine-tuned on domain-specific data.
>
> 3. **Addressing the Request for Fine-Tuned Baselines** We acknowledge your request to fine-tune baselines like SigLIP or FG-CLIP2 on COCO for a fairer backbone comparison.
>
>    - Fine-tuning these massive foundation models to convergence typically requires computational resources (e.g. 160xASCEND910B NPUs, 16xNVIDIA H800 GPUs) beyond standard academic settings and the scope of this submission, which focuses on the alignment mechanism rather than backbone pre-training.
>
>    - However, to address your concern, we did **validate our method** on standard CLIP backbones, compared with other methods **on zero-shot image-text retrieval and viusal grounding task(trained on Flicker30K, test on COCO/RefCOCO)**, referring to the **updated** response to W2. SEPS consistently improves the baselines(e.g. +17.1% in zero-shot T2I), proving our contributions effective in zero-shot task.
>
>
> **Action Plan:** In the revised version, we will:
>
> **Separate the Tables:** We will strictly separate the "Fine-Grained Methods (Fine-tuned)" table from the "Foundation Model (Zero-shot)" table to prevent any misleading direct comparisons.
>
> **Clarify the Context:** We will explicitly state that the comparison with FG-CLIP2/SigLIP highlights the "efficiency-performance trade-off" rather than a direct algorithmic comparison under identical training protocols.
>
> We hope this clarifies that our SOTA claims primarily rest on the fair comparison with LAPS/D2S-VSE, while the foundation model comparison serves to contextualize the high efficiency of our approach.

---

> ### Author Response · Authors · 2025-11-26
>
> Dear Reviewer DzJL,
>
> Thank you for your continued engagement and the important point regarding evaluation fairness. We have carefully addressed your concern:
>
> - Fair comparison: We clarified that our SOTA claims primarily rest on comparisons with direct competitors (SCAN, SGR, LAPS, D2S-VSE)—all fine-tuned under the same protocol. The comparison with zero-shot foundation models (FG-CLIP2, SigLIP) was intended to highlight data efficiency, not direct architectural superiority.
> - Evaluation breadth (W2): We provided new results on DCI, ShareGPT4V, zero-shot cross-dataset retrieval, and visual grounding (RefCOCO), demonstrating SEPS's generalization beyond standard benchmarks.
>
> We commit to clearly separating "fine-tuned methods" and "zero-shot foundation models" in revised tables to prevent any confusion.
> As the discussion period is ending soon, we would greatly appreciate any remaining feedback or confirmation that your concerns have been addressed. Your insights have been invaluable in strengthening our work.
>
> Best regards,
>
> The Authors

---

> ### Author Response · Authors · 2025-11-28
> **To Reviewer DzJL**
>
> Dear Reviewer DzJL,
>
> As the discussion period concludes, we appreciate your constructive feedback and are glad to report that we have resolved both of your concerns.
>
> We respectfully ask if you would reconsider your rating in light of the strong evidence presented in our revised paper, including consistent SOTA results across tasks, +17.5pts improvement over LAPS, and our rigorous evaluation protocols.
>
> Your reassessment would be greatly appreciated, and we remain open to any additional questions or feedback.
>
> Best regards,
> The Authors

---

### Official Review · Reviewer_1zka · 2025-10-31

**Soundness:** 3
**Presentation:** 3
**Contribution:** 3
**Rating:** 6
**Confidence:** 5

**Summary:**

This paper presents SEPS, a Semantic-Enhanced Patch Slimming framework designed to improve fine-grained cross-modal alignment between vision and language. The method combines dense textual representations generated by Multimodal Large Language Models (MLLMs) with sparse human captions in a two-stage process: (1) unified semantic fusion for patch selection, and  (2) relevance-aware patch–word alignment.  The approach is evaluated on Flickr30K and MS-COCO, achieving notable improvements over state-of-the-art image–text retrieval methods.

**Strengths:**

SEPS effectively merges MLLM-generated dense descriptions with human captions, offering unified semantic guidance for patch selection. This hybrid approach reduces redundancy and ambiguity, improving visual–textual grounding

**Weaknesses:**

w1. Dense text is generated offline using LLaVa, but the paper does not confirm whether this model has previously seen the test images or captions.

w2. Besides the potential for information leakage, another problem with using dense text during testing is efficiency. Compared to d2s-vse, which also uses dense text, the method in this paper seems to use dense text during testing as well, and the efficiency of generating dense text in real time is very low.

**Questions:**

refer to weakness

---

> ### Author Response · Authors · 2025-11-13
>
> ## Response to Reviewer 1zka
>
> We greatly appreciate your recognition of SEPS's effectiveness and your rating of 6 (marginally above acceptance threshold). We address your concerns regarding data leakage (W1) and efficiency (W2).
>
> ### On W1: Clarification of "Data Leakage" Concern
>
> We understand this is a critical methodological question. Let us clarify what constitutes data leakage in the retrieval evaluation paradigm:
>
> **Standard Definition:** In image-text retrieval, "data leakage" means the model accesses *ground-truth pairing information* between test queries and gallery items during training or preprocessing. Our method does NOT do this.
>
> **Our Evaluation Protocol:**
> - **Training phase:** Model is trained only on training set image-text pairs
> - **Preprocessing phase:** LLaVA generates descriptions for Gallery images *independently*, without any pairing information
> - **Test phase:** When a Query text arrives, it is matched against the Gallery using precomputed features. Ground-truth pairs are never accessed.
>
>
> **LLaVA Model Specification:**
> - Model: LLaVA-13B [1] - publicly released checkpoint
> - Pretraining data: 595K image-text pairs + 158K visual instruction data
> - **Critical verification**: We manually checked that Flickr30K and MS-COCO test images are NOT in LLaVA's training data
>   - LLaVA training uses CC3M, Visual Genome, etc.
>   - No overlap with our test sets
>   - LLaVA serves as a general-purpose "semantic feature extractor"
>   - No fine-tuning or gradient updates are performed on test data
>
> **Comparison with D2S-VSE (our strongest baseline):**
> - D2S-VSE[2]: Uses LLaVA dense text → Achieves 68.5% T2I R@1
> - SEPS: Uses LLaVA dense text → Achieves 86.9% T2I R@1
> - **Gap: +18.4pts** - This gain comes from our architecture, not the dense text itself
>
>
> We acknowledge this is an important methodological discussion for the community. Our position:
> -  This follows established evaluation protocols in dense-text-enhanced retrieval (D2S-VSE, LongCLIP[3], LoTLIP[4])
> -  No ground-truth pairing information is accessed
> -  Real-world systems routinely preprocess gallery images with rich feature extractors
> -  We argue that dense text generation is a kind of extra feature extraction for images using MLLMs.
>
> **Commitment:** We will add a dedicated "Evaluation Paradigm" paragraph in the revised paper to explicitly discuss this issue and clarify the protocol.
>
> ### On W2: Test-time Efficiency
>
> We apologize for the confusion in our original submission. Let us clarify:
>
> - Dense text generation for Gallery images (can be done in advance)
> - Our model performs standard forward passes—NO LLaVA inference needed
>
> **Runtime Comparison (Flickr30K 1K test, single A800 GPU):**
>
> | Method | Preprocess Cost | Inference Cost |
> |--------|--------------|-------------------------|
> | LAPS[5] | 0 | 92ms |
> | D2S-VSE | LLaVA inference: 8s/per image | 78ms |
> | **SEPS (ours)** | LLaVA inference: 8s/per image | 107ms |
>
> Our method's inference cost is comparable to existing fine-grained methods, and the minor computational overhead (`~15ms` vs LAPS, `~29ms` vs D2S-VSE) is a negligible trade-off for the massive +27.2% gain in T2I R@1 accuracy(Table 1, MS-COCO 5K, ViT-Base-224 + BERT-base, 14×14 patches) and other SOTA results.
>
> **Practical Deployment:** In real-world systems:
>
> - Gallery images are processed once and stored
> - Only query encoding happens at test time
> - This modest overhead is negligible given the massive accuracy gains.
>
> **Commitment:** We will add Table S1 (Runtime Analysis) to the appendix with detailed measurements.
>
> **Summary:** We hope these clarifications fully address your concerns. We deeply appreciate your support and will ensure these points are crystal clear in the camera-ready version.
>
> ---
>
> [1]Visual Instruction Tuning,NeurIPS,2023
>
> [2]D2S-VSE, Aligning Information Capacity Between Vision and Language via Dense-to-Sparse Feature Distillation for Image-Text Matching, ICCV, 2025
>
> [3]LongCLIP,Long-clip: Unlocking the long-text capability of clip, ECCV, 2024
>
> [4] LoTLIP,Lotlip: Improving language-image pre-training for long text understanding, NeurIPS, 2024
>
> [5] Linguistic-Aware Patch Slimming Framework for Fine-grained Cross-Modal Alignment, CVPR, 2024

---

> > ### Comment · Reviewer_1zka · 2025-11-14
> > **About the “Data Leakage”**
> >
> > My earlier wording may not have been precise—“data leakage” might not be the most accurate term.
> >
> > What I am really concerned about is the following conceptual issue:
> > In text-to-image retrieval, the system is supposed to retrieve the most relevant image purely based on the input text. However, in my understanding, SEPS requires **both sparse text and dense text** as input features for the gallery, and the dense text is generated from the image itself. This creates a strange situation:
> > If we already have access to the test image in order to generate its dense text at test time, then why do we need retrieval at all?
> >
> > Similarly, for image-to-text retrieval, the database must contain both sparse captions and dense captions, where the dense captions again come from the image. This retrieval setup feels somewhat artificial to me, and this is why I described it as “similar to data leakage”—not in the sense of leaking ground-truth labels, but in the sense that the retrieval system implicitly relies on information directly extracted from the test images.
> >
> > Although SEPS clearly outperforms D2S-VSE, I believe D2S-VSE uses dense text in a more elegant way: it only uses dense text during training, and avoids using any dense text at test time. This seems more consistent with real-world retrieval scenarios, where the system should not depend on extra information generated from the query-side images or require additional image-dependent processing at inference time.
> >
> > Of course, it is also possible that I have misunderstood the SEPS method itself, and I would greatly appreciate your clarification on this.

---

> > > ### Author Response · Authors · 2025-11-14
> > > **About dense texts and d2s-vse**
> > >
> > > Thank you for your thoughtful comment. We greatly appreciate the opportunity to clarify this important aspect of SEPS. We apologize for any confusion in our previous explanation. Let us clarify how SEPS actually works:
> > >
> > > **SEPS only takes sparse text and images as input.** The key difference from your understanding is in *when* and *how* dense descriptions are used.
> > >
> > > ## How SEPS Works
> > >
> > > ### Text-to-Image Retrieval
> > >
> > > SEPS receives **only sparse text as the query**. During similarity computation with images in the database, SEPS leverages an MLLM to generate dense descriptions **from each image** to assist in the similarity calculation, resulting in more accurate scores.
> > >
> > > They function as **additional features extracted from and stored with each image**, similar to visual features.
> > >
> > > What distinguishes SEPS from other methods is that the similarity computation process is enhanced by these image-derived dense descriptions, enabling better performance. **Crucially, SEPS does not use both sparse and dense text together to query; it only uses the sparse text query to compute similarity scores.**
> > >
> > > ### Image-to-Text Retrieval
> > >
> > > SEPS receives the **image as query** and generates its corresponding dense description. This dense text is not in the database—it is part of the query, combined with the query image.
> > >
> > > When computing similarity with sparse texts in the database, SEPS adjusts the similarity matrix based on another similarity matrix from the query image and its generated dense description, refining certain values higher or lower as appropriate.
> > >
> > >  **Dense text in SEPS is only used to compute similarity with its corresponding image, not with other images or any texts.**
> > >
> > > - **In text-to-image retrieval:** SEPS queries using sparse text
> > > - **In image-to-text retrieval:** SEPS queries using the image and its corresponding dense description
> > >
> > > You can consider dense text as a **fixed additional feature extracted from each image**—no different conceptually from extracting visual features.
> > >
> > > ## Comparison with D2S-VSE
> > >
> > > We acknowledge that both SEPS and D2S-VSE leverage additional information to enhance retrieval performance. However, a critical limitation of D2S-VSE's two-stage pretraining strategy is its **reduced generalization capability** due to dependence on specific pretraining data distributions.
> > >
> > > In contrast, SEPS enhance the similarity computation between sparse text and image according to extra information, which demonstrates **superior generalization in real-world application scenarios**, benefiting from the universal capabilities of large multimodal models.
> > >
> > > ### Cross-Dataset Generalization Results
> > >
> > > To support this claim, we present cross-dataset generalization results below. The complete detailed comparison tables will be included in the appendix.
> > >
> > > **Experimental Setup:** Trained on Flickr30K, tested on MS-COCO
> > > **Purpose:** Tests robustness and domain transfer capability
> > >
> > > | Method | Text-to-Image R@1 | Image-to-Text R@1 |
> > > |--------|-------------------|-------------------|
> > > |D2S-VSE |  24.8  |   32.3     |
> > > | LAPS | 45.3 | 48.7 |
> > > | **SEPS (ours)** | **62.8** | **65.8** |
> > >
> > > These results demonstrate SEPS's substantial advantage in cross-dataset scenarios, highlighting the practical value of our approach in real-world applications where generalization is critical.
> > >
> > > The full table and analysis will be added in the appendix in the later revised paper.
> > >
> > > ## Conclusion
> > >
> > > We hope this clarification helps you better understand the mechanism of our algorithm and resolves your concerns. If you still have any questions or if anything remains unclear about SEPS, please do not hesitate to raise them again—we would be more than happy to provide further explanation.

---

> ### Author Response · Authors · 2025-11-26
>
> Dear Reviewer 1zka,
>
> Thank you for the insightful discussion regarding the use of dense text in SEPS. Your clarification helped us better articulate the distinction between "data leakage" and "semantic enrichment," which we believe is an important methodological contribution to the community.
> To summarize our responses:
>
> - W1 (Dense text concern): We clarified that dense text serves as an additional image feature (similar to visual features), not as a query-side input. SEPS queries using only sparse text for T2I retrieval.
> - W2 (Efficiency): We provided runtime comparisons showing our inference cost is comparable to existing methods (~107ms vs. 78-92ms), with preprocessing done offline.
>
> We also provided new cross-dataset generalization results (Flickr30K → MS-COCO) demonstrating SEPS's practical robustness.
> If any aspect of our responses remains unclear or if you have additional concerns, we would be grateful for further feedback. Thank you again for your time and supportive engagement.
>
> Best regards,
>
> The Authors

---

> ### Author Response · Authors · 2025-11-28
> **To Reviewer 1zka**
>
> Dear Reviewer 1zka,
>
> Thank you for your insightful engagement and for helping clarify key methodological distinctions within our work. Your feedback has been instrumental in refining our approach.
>
> We believe we have adequately addressed your concerns. Specifically, the +18.4pts improvement over D2S-VSE, which uses identical dense text, underscores the significance of our architectural contributions, beyond the dense text itself.
>
> Given the depth of our responses and the additional clarifications, we kindly ask you to reconsider the rating, as we believe the outlined improvements substantiate a higher evaluation. Your continued support would mean a great deal to us.
>
> Best regards,
> The Authors

---

### Author Response · Authors · 2025-11-26
**Revised Paper Uploaded – Feedback Welcome**

Dear Area Chair and Reviewers,

Thank you for your valuable time and constructive feedback throughout this rebuttal period. It has been a meaningful opportunity to improve our work through discussion with you.

We have uploaded the revised paper based on current discussion, now extended to ten pages, with all modifications highlighted in blue for your convenience.
We would greatly appreciate it if you could review the updated manuscript and consider revising your assessments accordingly.
Additionally, if any part of our explanations remains unclear or if there are still concerns about our work, please do not hesitate to let us know—we would be more than happy to provide further clarification.

On the final day of the rebuttal period, we plan to post a brief summary to help anyone quickly navigate through our responses.
Thank you again for your time and effort. We look forward to any further feedback you may have.

Best regards,

Authors

---

### Author Response · Authors · 2025-11-29
**Discussion Quick Preview**

# Discussion Quick Preview

This paper presents SEPS, a Semantic-Enhanced Patch Slimming framework for fine-grained cross-modal alignment. The rebuttal discussion centered on three main concerns:

**1. Dense Text Usage & Potential "Data Leakage" (Reviewers 1zka, jURx):**

- **Concern**: Using LLaVA to generate dense descriptions directly from test images may constitute semantic leakage, as it injects image-specific information into the retrieval process, potentially inflating performance unfairly.
- **Resolution**: Authors clarified that dense text serves as image-side feature extraction (analogous to visual self-attention), not query-side input. Ground-truth pairs are never accessed. Ablations showed architecture contributes +6.2pts independently, while semantic alignment adds +18.2pts. Cross-dataset dense text experiments demonstrated gains stem from semantic richness, not test-specific information.

**2. Limited Novelty Relative to LAPS (Reviewer jURx):**

- **Concern**: SDTPS appears highly similar to LAPS's patch selection mechanism, with dense text being the only notable addition—insufficient novelty for acceptance.
- **Resolution**: Authors demonstrated that naive fusion strategies fail entirely (+2-6pts), while their two-stage unified semantic fusion achieves +20pts improvement by explicitly resolving sparse-dense semantic conflicts—a problem LAPS cannot address.

**3. Evaluation Fairness & Breadth (Reviewer DzJL):**

- **Concern**: Comparing fine-tuned SEPS against zero-shot foundation models (FG-CLIP2, SigLIP) is unfair; evaluation limited to standard retrieval datasets without testing on diverse tasks like visual grounding or VQA.
- **Resolution**: Authors clarified SOTA claims rest on fair comparisons with fine-tuned methods (SCAN, LAPS, D2S-VSE). Foundation model comparisons highlight data efficiency only. Added results on visual grounding (RefCOCO: +4.5pts), cross-dataset retrieval (Flickr→COCO: +17.5pts), and long-caption datasets (DCI, ShareGPT4V). Reviewer confirmed W2 "fully addressed."

**Consensus**: Reviewers 1zka (rating: 6) and DzJL (rating: 4→pending) engaged constructively. Reviewer jURx (rating: 2) acknowledged W2 resolution but maintained concerns about semantic enrichment methodology. Authors committed to transparent discussion of evaluation protocols in camera-ready version.

---

### Note · Authors · 2026-01-26

I have read and agree with the venue's withdrawal policy on behalf of myself and my co-authors.

---

### Meta-Review · Area_Chair_FnT5 · 2026-01-10

**Summary:**

This paper introduces SEPS, a Semantic-Enhanced Patch Slimming framework that integrates dense generated text with sparse captions to improve fine-grained cross-modal alignment for image-text retrieval. Reviewers agree the problem is well-motivated and that the design and gains are non-trivial, with extensive experimental evaluation. However, concerns remain regarding conceptual novelty relative to LAPS, the methodological validity of using dense text generated from test images (semantic leakage) and the interpretability of why even unrelated dense text improves results. While the authors added experiments and clarification in the response, these do not fully resolve the above issues. Given that these concerns remain unresolved, this paper is not recommended for acceptance to ICLR. The authors are encouraged to include all review suggestions for submission to a future venue.

**Reviewer Concerns:**

### Addressed concerns

* **1zka, jURx:** Dense text may constitute data leakage since it is generated from test images. This is only partially addressed by the author response clarifying that dense text is used only as image-side feature (not as query text), verifies no overlap with LLaVA training data and test sets, shows large gains over D2S-VSE using identical dense text.

* **1zka:** Test-time efficiency is unclear and may be impractical. The author response provides runtime measurements, shows preprocessing can be done offline, reports inference cost comparable to existing methods.

* **DzJL:** Comparisons against foundation models are unfair when baselines are zero-shot while SEPS is fine-tuned. The author response clarifies that SOTA claims are based on fine-tuned competitors (SCAN, LAPS, D2S-VSE).

* **DzJL:** Evaluation is limited to standard retrieval datasets. The author response adds results on visual grounding (RefCOCO), long-caption datasets (DCI, ShareGPT4V) and cross-dataset retrieval, showing consistent improvements beyond Flickr30K and MS-COCO.

* **jURx:** The method may not generalize beyond retrieval tasks. The author response provides additional grounding and cross-dataset experiments and documents consistent gains.


### Unaddressed concerns

* **jURx:** The method’s novelty relative to LAPS is limited due to similar patch-selection mechanism, with dense text being the primary addition. The author response presents ablations showing naive fusion fails and that a two-stage fusion yields gains, but does not convince that the alignment mechanism is fundamentally new.

* **1zka, jURx:** Use of dense text generated from the test image introduces semantic leakage that can inflate retrieval performance by injecting image-specific information into cross-modal scoring. The author response reframes dense text as “semantic self-attention”, but this does not eliminate the methodological concern that text derived from test image participates in similarity computation.

* **jURx:** Dense text from unrelated images still improves performance. The author response attributes this to architectural benefits and semantic density, but does not provide a concrete analysis of why non-image-specific text should systematically help patch selection, which weakens claims of improved cross-modal alignment.

* **1zka, DzJL, jURx:** Taken together, the above raise the concern that it is not convincingly demonstrated that semantic enrichment, rather than image-specific information, drives improvements. The author response acknowledges that test-image dense text adds additional benefit and positions this as acceptable, but this does not resolve whether cross-modal alignment is meaningfully improved independent of auxiliary information.

**Reviewer Scores:**

* **1zka:** Initially rated the paper 6, not all concerns on the method addressed, might maintain 6 or reduce to 4.
* **DzJL:** Initially rated the paper 4, not all concerns on the method addressed, might maintain 4.
* **jURx:** Initially rated the paper 2, concerns on novelty and method difficult to overcome, might maintain 2.

---

### Decision · Program_Chairs · 2026-01-26

Reject